# Riemannian Stochastic Weakly Convex Optimization Under Heavy-Tailed Noises

## Abstract

Stochastic optimization in the presence of heavy-tailed noise has been extensively studied in Euclidean spaces, yet remains poorly understood in non-Euclidean geometry. Many modern learning problems, such as representation learning and optimization with orthogonality constraints, naturally involve optimization on Riemannian manifolds where gradient noise can exhibit heavy-tailed behavior. In this work, we establish the first high-probability convergence guarantees for Riemannian stochastic subgradient descent (SsGD) under weakly convex objectives with sub-Weibull-type noise, as well as both high-probability and in-expectation convergence guarantees for a Riemannian clipped-SsGD under noise with only a bounded $p$-th central moment ($p \in (1, 2]$). Our analysis shows that, despite the geometric and statistical difficulties, the dependence on failure probability and iteration complexity matches the best-known Euclidean results up to constants. These results highlight that manifold geometry and weak convexity do not incur additional penalties under heavy-tailed stochasticity.

## 1 Introduction

In the paper, we consider the stochastic optimization problem on Riemannian manifolds

$$\min_x \mathbb{E}_{\zeta \sim D}[f(x, \zeta)] \quad \text{subject to } x \in \text{St}(n, r), \tag{1}$$

where the data $\zeta$ is a random variable following an unkonwn distribution $D$, and the function $f$ is $\rho$-weakly convex, meaning that for some constant $\rho \geq 0$, the assignment $f(\cdot) + \frac{\rho}{2}\|x\|^2$ is convex. In particular, the objective function in problem (1) can be nonsmooth and nonconvex. The problem (1) has a wide range of applications, including principal component analysis (PCA) Wold et al. (1987), dictionary learning Cherian & Sra (2016); Sun et al. (2016), Gaussian mixture models Hosseini & Sra (2015), low-rank matrix/tensor completion Vandereycken (2013); Kasai & Mishra (2016), low-rank multivariate regression Meyer et al. (2011), so on.

Stochastic optimization algorithms have been extensively studied to solve problems on Riemannian manifolds. The Riemannian stochastic gradient descent (SGD) method was proposed by Bonnabel (2013). Other notable works include Zhang et al. (2016); Huang et al. (2018); Wang et al. (2022); Sakai & Iiduka (2023), among others. These methods have demonstrated remarkable effectiveness in solving optimization problems on Riemannian manifolds. Specifically, in the nonsmooth and weakly convex setting, Li et al. (2021) proposed a Riemannian stochastic subgradient algorithm with an iteration complexity of $O(\varepsilon^{-4})$. Davis et al. (2025) introduced a class of stochastic algorithms designed to minimize weakly convex functions over proximally smooth sets. A critical assumption in their theoretical analysis is that the gradient noise is assumed to be bounded in variance, which is defined as a light-tailed distribution in statistical terms.

A critical challenge in SGD is the inherent noise in gradient estimation. While existing work often assumes bounded noise variance Yu et al. (2019); Yi et al. (2022); Assran et al. (2019), this premise is often unrealistic; heavy-tailed noise is more common in practice. Studies Gurbuzbalaban et al. (2021); Armacki et al. (2023); Garg et al. (2021); Battash et al. (2024) show that gradient noise frequently follows a heavy-tailed distribution, violating the finite-variance assumption. Under such heavy-tailed noise, SGD often performs poorly and may even diverge. Some recent work in Euclidean spaces—such as that of Zhu et al. (2025)—has shown that vanilla stochastic subgradient descent achieves high-probability convergence under sub-Weibull noise, while its clipped variant

also achieves both high-probability and in-expectation convergence under $p$-BCM noise. Their research focuses on constrained nonsmooth nonconvex optimization problems, where the constraint set is closed and convex while the objective function is nonsmooth. This represents a significant advancement. Moreover, recent empirical observations suggest that gradient noise on Riemannian manifolds may follow a heavy-tailed distribution. For example, Brunnström et al. (2024) transformed the problem of signal and noise covariance matrices estimation into an unconstrained optimization problem on a Riemannian manifold, derived a robust covariance matrix estimator based on Tyler's M-estimator, and demonstrated its effectiveness under heavy-tailed noise. However, this work lacks a theoretical analysis on Riemannian manifolds under heavy-tailed noise conditions. These observations therefore motivate us to extend the approach of Zhu et al. (2025) to Problem (1) on Riemannian manifolds, proposing that the assumptions on gradient noise over manifolds should be strictly weaker than the bounded variance condition. In this problem, nonconvex constraint sets are typically characterized by Riemannian manifolds, and the objective function is both nonconvex and nonsmooth.

## 1.1 MAIN CONTRIBUTIONS

This paper aims to provide a convergence analysis of stochastic first-order methods for solving Problem (1) under heavy-tailed noise in the context of nonsmooth and weakly convex optimization on Riemannian manifolds. The main contributions of this work are summarized as follows.

- We establish the first high-probability convergence rate for Riemannian SsGD on the Stiefel manifold under nonsmooth weak convex optimization with sub-Weibull type noise, which is $O\left((\log(T/\delta)^{\min\{0,\theta-1\}} \log(1/\delta) + \rho \log(1/\delta)^{2\theta} \log T)/T\right)$ for an unknown time horizon $T$ and $O\left(\sqrt{(\rho \log(1/\delta)^{2\theta})/T} + \left(\log(T/\delta)^{\min\{0,\theta-1\}} \log(1/\delta)\right)/T\right)$ for a fixed time horizon $T$. The results indicate that the dependence of the number of iterations and the failure probability $\delta$ aligns with existing works in Euclidean settings, demonstrating that the theoretical convergence rate of Riemannian SsGD for nonconvex, nonsmooth yet weakly convex objectives on the Stiefel manifold does not deteriorate compared to its nonsmooth weakly convex counterpart in Euclidean space under sub-Weibull noise.

- We establish the first high-probability convergence rate for mini-batch Riemannian clipped-SsGD on the Stiefel manifold under nonsmooth weakly convex optimization with $p$-BCM type noise. The dependence on the failure probability $\delta$ is $O(\log(T/\delta))$, consistent with existing results for nonsmooth optimization in Euclidean settings, while the sample complexity is $O(\varepsilon^{-2p/(p-1)})$. Notably, in our analysis, the logarithmic term $\log T$ can be eliminated by fixing the time horizon $T$ for high-probability convergence.

## 2 RELATED WORK

### 2.1 NONSMOOTH WEAKLY CONVEX STOCHASTIC OPTIMIZATION IN $\mathbb{R}^n$

The problem of minimizing a weakly convex function on the convex constraint set $\mathbb{R}^n$ has been thoroughly studied in the literature. The main algorithms for solving this task include subgradient-type methods Davis & Drusvyatskiy (2018; 2019); Li et al. (2019) and proximal point-type methods Drusvyatskiy (2017). Furthermore, there have also been some recent research advancements in the field of nonsmooth weakly convex stochastic optimization. Chen et al. (2021) proposed a distributed stochastic subgradient method for solving weakly convex problems over networks. Pougkakiotis & Kalogerias (2023) proposed a zeroth-order proximal stochastic gradient method designed for nonsmooth and weakly convex stochastic composite optimization problems. Hu et al. (2023) introduced a single-loop stochastic optimization algorithm aimed at solving nonsmooth weakly convex finite-sum coupled compositional optimization problems. Gao & Deng (2023) introduced a delayed stochastic approximation minimization (DSPL) method to address an important class of composite weakly convex problems. Yang et al. (2025) proposed a single-loop penalty-based stochastic algorithm to address finite-sum coupled compositional problems, where both the objective and constraint functions are weakly convex.

## 2.2 NONSMOOTH WEAKLY CONVEX STOCHASTIC OPTIMIZATION OVER RIEMANNIAN MANIFOLD

Recently, some works have been proposed to address the nonsmooth weakly convex optimization problems on Riemannian manifolds. For instance, Li et al. (2021) focused on a class of nonsmooth weakly convex optimization problems on the Stiefel manifold. They proposed a Riemannian stochastic subgradient type algorithms and proved that it has an iteration complexity of $O(\varepsilon^{-4})$ for driving a natural stationary measure below $\varepsilon$. Peng et al. (2023) proposed a Riemannian smoothing stochastic gradient method and gave an iteration complexity of $O(\varepsilon^{-3})$ for driving a generalized $\varepsilon$-stationary point. Davis et al. (2025) introduced a class of stochastic algorithms for minimizing weakly convex functions over proximally smooth sets. Note that a key assumption in their theoretical analysis is that the gradient noise is variance bounded, which is defined as a light-tailed distribution.

## 2.3 STOCHASTIC OPTIMIZATION UNDER HEAVY-TAILED NOISES

Madden et al. (2020) pioneered the high-probability convergence analysis of SGD for smooth optimization under sub-Weibull type noises. Subsequent works, including Li & Liu (2022), Liu et al. (2022), Liu & Zhou (2023a), Li et al. (2025), and Yu et al. (2025), extended this line of research by providing high-probability or in-expectation convergence guarantees for various stochastic gradient-based methods in both convex and smooth optimization settings with sub-Weibull noises. However, as far as we know, no existing work has investigated the high-probability convergence of ordinary SGD methods under sub-Weibull noises for nonsmooth and weakly convex optimization on Riemannian manifolds.

Zhang et al. (2020) were the first to put forward the in-expectation upper bounds for clipped-SGD under $p$-BCM noises. Subsequently, Cutkosky & Mehta (2021), Nguyen et al. (2023), Liu et al. (2023b), Sadiev et al. (2023), Liu & Zhou (2023b) and Chezhegov et al. (2025) also conducted high-probability or in-expectation convergence analysis for various clipped-SGD methods under $p$-BCM noises. Closely aligned with our research, Liu et al. (2024) introduced a first-order online optimization algorithm that ensures high-probability convergence under $p$-BCM noises in nonsmooth, non-convex online settings. Although their algorithm framework accommodates nonsmooth objectives, it still necessitates continuous differentiability. Hu et al. (2025) established the first in-expectation convergence guarantees for clipped-SGD under $p$-BCM noises in weakly convex optimization. Zhu et al. (2025) established both high-probability and in-expectation convergence guarantees for clipped SsGD under $p$-BCM noise for weakly convex optimization. Nevertheless, to the best of our knowledge, no existing work has studied the high-probability convergence guarantees of clipped-SGD under $p$-BCM noises for weakly convex optimization on Riemannian manifolds.

## 2.4 NOTATIONS

For any $N > 1$, We define $[N]$ as the set $\{1, \cdots, N\}$. Let $\mathrm{St}(n, r)$ be a Stiefel manifold, and we use $T_x\mathrm{St}(n, r) := \{\xi \in \mathbb{R}^{n \times r} : \xi^\top x + x^\top \xi = 0\}$ to denote the tangent space of $\mathrm{St}(n, r)$ at the point $x \in \mathrm{St}(n, r)$. Give two matrices $A$ and $B$, we use $\langle A, B \rangle = \mathrm{trace}(A^\top B)$ denote the Euclidean inner product. For a close set $\mathcal{C} \subseteq \mathbb{R}^{n \times r}$, let $P_\mathcal{C}$ denote the orthogonal projector onto $\mathcal{C}$, and define the distance from $x$ to $\mathcal{C}$ as $\mathrm{dist}(x, \mathcal{C}) := \inf_{y \in \mathcal{C}} \|x - y\|$.

## 3 PRELIMINARIES

The Riemannian subdifferential of $f$ on the Stiefel manifold $\mathrm{St}(n, r)$ is given by

$$\partial_\mathcal{R} f(x) = \mathcal{P}_{T_x\mathrm{St}(n,r)}\left(\partial f(x)\right), \quad \forall x \in T_x\mathrm{St}(n, r). \tag{2}$$

In particular, given an Euclidean subgradient $\widetilde{\nabla} f(x) \in \partial f(x)$ of $f$ at $x \in \mathrm{St}(n, r)$, we get a corresponding Riemannian subgradient $\widetilde{\nabla}_\mathcal{R} f(x) \in \partial_\mathcal{R} f(x)$ through $\widetilde{\nabla}_\mathcal{R} f(x) = \mathcal{P}_{T_x\mathrm{St}(n,r)}(\widetilde{\nabla} f(x))$.

Applying (2), we say $x \in \mathrm{St}(n, r)$ is a stationary point if it satisfies the following first-order optimality condition:

$$0 \in \partial_\mathcal{R} f(x). \tag{3}$$

For any $\lambda > 0$, the Moreau envelope and the proximal mapping are defined as

$$
\begin{cases}
f_\lambda(x) = \min_{y \in \mathrm{St}(n,r)} \left\{ f(y) + \dfrac{1}{2\lambda} \|y - x\|^2 \right\}, \\[2mm]
P_{\lambda f}(x) = \arg\min_{y \in \mathrm{St}(n,r)} \left\{ f(y) + \dfrac{1}{2\lambda} \|y - x\|^2 \right\}.
\end{cases}
\tag{4}
$$

According to (2) and (3), the proximal point $P_{\lambda f}(x)$ satisfies the first-order optimality condition, i.e., $0 \in \partial_\mathcal{R} f\left(P_{\lambda f}(x)\right) + \frac{1}{\lambda} \mathcal{P}_{T_{P_{\lambda f}(x)} \mathrm{St}(n,r)}\left(P_{\lambda f}(x) - x\right)$. Then we can obtain

$$
\mathrm{dist}\left(0, \partial_\mathcal{R} f\left(P_{\lambda f}(x)\right)\right) \leq \lambda^{-1} \cdot \left\| \mathcal{P}_{T_{P_{\lambda f}(x)} \mathrm{St}(n,r)}\left(P_{\lambda f}(x) - x\right) \right\|
$$
$$
\leq \lambda^{-1} \cdot \|P_{\lambda f}(x) - x\| =: \Theta(x).
\tag{5}
$$

At the current iterate $x_t$, the Riemannian stochastic subgradient oracle generates a sample $\zeta_t \sim D$ that is independent of $\{\zeta_0, \dots, \zeta_{t-1}\}$ and returns a Riemannian stochastic subgradient $\tilde{\nabla}_\mathcal{R} f(x_t, \zeta_t)$. Then, the Riemannian stochastic subgradient method generates the next iterate $x_{t+1}$ via

$$
x_{t+1} = \mathcal{R}_{x_t}(\xi_t) \quad \text{with} \quad \xi_t = -\eta_t \tilde{\nabla}_\mathcal{R} f(x_t, \zeta_t).
\tag{6}
$$

where $\eta_t > 0$ is the step size, $\mathcal{R}$ is any retraction on the Stiefel manifold, see (Absil et al., 2008, Setion 4.2).

**Assumption 3.1.** *The function $f$ is $\rho$-weakly convex with $\rho > 0$;*

**Assumption 3.2.** *The function $f(x)$ attains at least one local minimum $f^*$ on $\mathrm{St}(n,r)$ and $f^* > -\infty$;*

**Assumption 3.3.** *There exists $G > 0$, for any $x \in \mathrm{St}(n,r)$ and any $\tilde{\nabla}_\mathcal{R} f(x) \in \partial_\mathcal{R} f(x)$, we have $\|\tilde{\nabla}_\mathcal{R} f(x)\| \leq G$;*

**Assumption 3.4.** *The stochastic Riemannian gradient oracle is unbiased, satisfying $\mathbb{E}_\zeta[\tilde{\nabla}_\mathcal{R} f(x, \zeta)] \in \partial_\mathcal{R} f(x)$.*

**Assumption 3.5.** *The norm of gradient noise $\left\| \tilde{\nabla}_\mathcal{R} f(x, \zeta) - \mathbb{E}_\zeta[\tilde{\nabla}_\mathcal{R} f(x, \zeta)] \right\|$ follows a $\sigma$-sub-Weibull($\theta$) distribution with $\theta \geq 1/2$;*

**Assumption 3.6.** *The gradient noise $\tilde{\nabla}_\mathcal{R} f(x, \zeta) - \mathbb{E}_\zeta[\tilde{\nabla}_\mathcal{R} f(x, \zeta)]$ satisfies $p$-BCM condition for $p \in (1, 2]$.*

**Remark 3.1.** *Assumptions 3.2 and 3.4 are the standard conditions in non-smooth stochastic optimization on Riemannian manifolds, while assumption 3.3 is formulated to ensure that the Riemannian subgradient has an upper bound.*

## 4 MAIN RESULTS

In this section, we present the main theoretical results of this paper. Firstly, in Section 4.1 under the weakly convex setting of the Stiefel manifold, we analyze the high-probability convergence of the Riemannian stochastic subgradient descent (SsGD) method with sub-Weibull type gradient noise. Then, in Section 4.2, still within the weakly convex setting, we examine the high-probability convergence of Clipped-SsGD under $p$-th moment type noise, which permits the second moment norm of the gradient noise to be unbounded when $1 < p < 2$.

### 4.1 RIEMANNIAN SsGD METHOD UNDER THE SUB-WEIBULL NOISES

In this section, we formally conduct the convergence analysis of the SsGD method (Algorithm 1) for weakly convex optimization on the Stiefel manifold under $\sigma$-sub-Weibull($\theta$)-type noise. First of all, let's briefly introduce the sub-Weibull noise distribution.

A random vector $X$ is said to follow a $\sigma$-sub-Weibull($\theta$) distribution if there exists a nonnegative real number $\sigma$ satisfying

$$
\mathbb{E}\left[ \exp\left\{ (|X|/\sigma)^{1/\theta} \right\} \right] \leq 2,
\tag{7}
$$

with $\theta \geq 1/2$ denoting the tail parameter. The sub-Weibull family encompasses a broader class of distributions compared to the sub-Gaussian family. Specifically, the case $\theta = 1/2$ corresponds to sub-Gaussian distributions, while $\theta = 1$ yields sub-exponential distributions. Heavier tails are observed as the tail parameter $\theta$ increases. For more equivalent definitions of the sub-weibull distribution, interested readers can refer to Vladimirova et al. (2020). Previous studies, including Madden et al. (2020), Li & Liu (2022), and Li et al. (2025), have investigated the high-probability convergence of stochastic first-order methods for smooth optimization in Euclidean space under sub-Weibull noises.

Now, we present the Riemannian SsGD algorithm on the Stiefel manifold as follows.

---

**Algorithm 1** Riemannian SsGD method

---

1: Input: $x_0 \in \mathrm{St}(n, r)$ and $\eta_0 > 0$
2: **for** $t = 1, \cdots, T$ **do**
3:    Access to unbiased Riemannian stochastic gradient oracle to get $\tilde{\nabla}_{\mathcal{R}} f(x_t)$
4:    $\xi_t = -\eta_t \tilde{\nabla}_{\mathcal{R}} f(x_t)$
5:    $x_{t+1} = \mathcal{R}_{x_t}(\xi_t)$
6: **end for**

---

Then we present a general high-probability convergence theorem of the Riemannian SsGD method under sub-Weibull-type noise for weak convex optimization on the Stiefel manifold.

**Theorem 4.1.** *Suppose Assumption 3.1, 3.2, 3.3, 3.4 and 3.5 hold. Let $\{x_t\}$ denote the iterate generated by the Riemannian SsGD method. Define $f_{1/\bar{\rho}(x_t)}$ as in (4), and set $\bar{\rho} = 3(\rho + L)$, where $L$ is Lipschitz constant. If $\eta_t$ satisfy*

$$\sum_{t=1}^{T} \eta_t = +\infty, \quad \sum_{t=1}^{T} \eta_t^2 \leq +\infty,$$

*then for any $\delta \in (0, 1)$, the following holds with the probability at least $1 - \delta$ :*

- *when $\theta = \frac{1}{2}$, we get following inequality*

$$\sum_{t=1}^{T} \frac{\eta_t}{\sum_{s=1}^{T} \eta_s} \|\Theta(x_t)\|^2 \leq O \left( \frac{\Delta_1 + \sigma^2 \log(1/\delta) \max_{t \in [T]} \eta_t}{\sum_{t=1}^{T} \eta_t} \right.$$
$$\left. + \frac{\rho(\log(1/\delta)\sigma^2 + G^2) \sum_{t=1}^{T} \eta_t^2}{\sum_{t=1}^{T} \eta_t} \right) \qquad (8)$$

- *when $\theta \in \left( \frac{1}{2}, 1 \right]$, we get following inequality*

$$\sum_{t=1}^{T} \frac{\eta_t}{\sum_{s=1}^{T} \eta_s} \|\Theta(x_t)\|^2 \leq O \left( \frac{\Delta_1 + \max\{\sigma^2, G\sigma\} \log(1/\delta) \max_{t \in [T]} \eta_t}{\sum_{t=1}^{T} \eta_t} \right.$$
$$\left. + \frac{\rho((\log(1/\delta))^{2\theta}\sigma^2 + G^2) \sum_{t=1}^{T} \eta_t^2}{\sum_{t=1}^{T} \eta_t} \right) \qquad (9)$$

- *when $\theta > 1$, we get following inequality*

$$\sum_{t=1}^{T} \frac{\eta_t}{\sum_{s=1}^{T} \eta_s} \|\Theta(x_t)\|^2 \leq O \left( \frac{\Delta_1 + D(\theta) \log(1/\delta) \max_{t \in [T]} \eta_t}{\sum_{t=1}^{T} \eta_t} \right.$$
$$\left. + \frac{\rho((\theta \log(1/\delta))^{2\theta}\sigma^2 + G^2) \sum_{t=1}^{T} \eta_t^2}{\sum_{t=1}^{T} \eta_t} \right) \qquad (10)$$

*where $\Delta_1 \triangleq f_{1/3(\rho+L)}(x_1) - \min_{\mathrm{St}(n,r)} f(x)$, $D(\theta) = \max \left\{ G\sigma \left( \log(T/\delta) \right)^{\theta-1}, 2^{3\theta+1}\Gamma(3\theta + 1)\sigma^2 \right\}$ and $\Gamma(\cdot)$ is Gamma function.*

Note that Theorem 4.1 does not require any additional explicit formula for $\eta_t$. The detailed proof of Theorem 4.1 can be found in C.1 of the Appendix. Furthermore, in Theorem 4.1, certain constants in the big-O complexity of the upper bound have been omitted for brevity. Our analysis is applicable to any case where $\bar{\rho} > \rho$. In the proof, we set $\bar{\rho} = 3(\rho + L)$ merely to simplify the upper bound in Theorem 4.1.

From Theorem 4.1, we can observe that as $\theta$ increases, the tail of the gradient noise distribution becomes heavier, and the iteration complexity of Riemannian SsG deteriorates. This aligns with empirical observations of projected SsGD methods in Euclidean space when handling heavy-tailed gradient noise. Interestingly, for $\theta > 1$, although we do not specify an explicit form of $\eta_t$, an additional term $\log(1/\delta)(\log(T/\delta))^{\theta-1}$ arises in the complexity upper bound. In our proof, the appearance of $\log(1/\delta)(\log(T/\delta))^{\theta-1}$ stems from the use of Theorem A.1, which can be regarded as a generalization of of the sub-Gaussian freedman inequality in Theorem 1 of Li & Orabona (2020). It is worth noting that for $\theta > 1$, the highest order of $\sigma$ is 2, but the coefficient preceding $\sigma^2$ is the factorial function $\Gamma(3\theta + 1)$, which grows significantly as $\theta$ increases. All the above analysis demonstrates that Riemannian SsGD exhibits degraded performance in the presence of $\sigma$-sub-Weibull($\theta$)-type gradient noise when $\theta > 1$, even when the squared norm of the noise remains bounded.

By setting $\eta_t = O(1/\sqrt{t})$, the following corollary can be directly derived from Theorem 4.1.

**Corollary 4.1.** *Suppose Assumptions 3.1, 3.2, 3.3, 3.4 and 3.5 hold. Let $\{x_t\}$ denote the iterate generated by the Riemannian SsGD method. Define $f_{1/\bar{\rho}}(x_t)$ as in (4), and set $\bar{\rho} = 3(\rho + L)$, where $L$ is Lipschitz constant. Set $\eta_t = \frac{\gamma}{\sqrt{t}}$, where $\gamma$ is an arbitrary positive constant. For any $\delta \in (0, 1)$, the Riemannian SsGD method converges with the probability at least $1 - \delta$*

- *For $\theta = \frac{1}{2}$, the following inequality holds:*

$$\frac{1}{T}\sum_{t=1}^{T}\|\Theta(x_t)\|^2 \leq O\left(\frac{\Delta_1}{\sqrt{T}} + \rho\left(\log(1/\delta)\,\sigma^2 + G^2\right)\frac{\log(T)}{\sqrt{T}}\right) \tag{11}$$

- *For $\theta \in \left(\frac{1}{2}, 1\right]$, the following inequality holds:*

$$\frac{1}{T}\sum_{t=1}^{T}\|\Theta(x_t)\|^2 \leq O\left(\frac{\Delta_1 + \max\{\sigma^2, G\sigma\}\log(1/\delta)}{\sqrt{T}}\right.$$
$$\left. +\rho((\log(1/\delta))^{2\theta}\sigma^2 + G^2)\frac{\log(T)}{\sqrt{T}}\right) \tag{12}$$

- *For $\theta > 1$, the following inequality holds:*

$$\frac{1}{T}\sum_{t=1}^{T}\|\Theta(x_t)\|^2 \leq O\left(\frac{\Delta_1 + D(\theta)\log(1/\delta)}{\sqrt{T}} + \rho((\theta\log(1/\delta))^{2\theta}\sigma^2 + G^2)\frac{\log(T)}{\sqrt{T}}\right)$$
$$\tag{13}$$

*where $D(\theta)$ is consistent with the meaning in Theorem 4.1.*

Note that when $\sigma = 0$, for any value of $\theta$, the result of Corollary 4.1 can be simplified to a deterministic case, that is, $O\left((\Delta_1 + \rho G^2 \log T)/\sqrt{T}\right)$. Furthermore, in Corollary 4.1, a parameter-free and time-varying step-size is employed. A natural extension is to consider a fixed time-horizon $T$ and assume that the algorithm can obtain the problem parameters (such as $\Delta_1$, $G$, and $\sigma$) to improve the upper bound in Corollary 4.1.

**Corollary 4.2.** *Suppose Assumption 3.1, 3.2, 3.3, 3.4 and 3.5 hold. Let $\{x_t\}$ denote the iterate generated by the Riemannian SsGD method. Define $f_{1/\bar{\rho}}(x_t)$ as in (4), and set $\bar{\rho} = 3(\rho + L)$, where $L$ is Lipschitz constant. Fix the time-horizon $T$ and assume that the algorithm has access to the parameters $\Delta_1$, $G$, and $\sigma$. Then, for any $\delta \in (0, 1)$, the following holds with the probability at least $1 - \delta$*

- *For $\theta = \frac{1}{2}$, setting $\eta_t = \sqrt{\frac{3\Delta_1}{(\rho+L)(36e\log(4/\delta)\sigma^2+9G^2)T}}$ we get following inequality:*

$$\frac{1}{T}\sum_{t=1}^{T}\|\Theta(x_t)\|^2 \leq O\left(\sqrt{\frac{\rho\Delta_1\left(\log(1/\delta)\sigma^2+G^2\right)}{T}} + \frac{\log(1/\delta)\sigma^2}{T}\right) \quad (14)$$

- *For $\theta \in (\frac{1}{2}, 1]$, setting $\eta_t = \sqrt{\frac{3\Delta_1}{(\rho+L)(2130\log(4/\delta)^{2\theta}\sigma^2+9G^2)T}}$, we get following inequality:*

$$\frac{1}{T}\sum_{t=1}^{T}\|\Theta(x_t)\|^2 \leq O\left(\sqrt{\frac{\rho\Delta_1\left((\log(1/\delta))^{2\theta}\sigma^2+G^2\right)}{T}} + \frac{\max\{\sigma^2, G\sigma\}\log(1/\delta)}{T}\right) \quad (15)$$

- *For $\theta > 1$, setting $\eta_t = \sqrt{\frac{3\Delta_1}{(\rho+L)(18(11\theta\log(4/\delta))^{2\theta}\sigma^2+9G^2)T}}$, we get following inequality:*

$$\frac{1}{T}\sum_{t=1}^{T}\|\Theta(x_t)\|^2 \leq O\left(\sqrt{\frac{\rho\Delta_1\left((\theta\log(1/\delta))^{2\theta}\sigma^2+G^2\right)}{T}} + \frac{D(\theta)\log(1/\delta)}{T}\right) \quad (16)$$

Note that where $\sigma = 0$, for any value of $\theta$, the result of Corollary 4.2 can be simplified to a deterministic case, that is, $O\left(\sqrt{(\rho\Delta_1 G^2)/T}\right)$ with $\eta_t = O\left(\sqrt{\Delta_1/(\rho G^2 T)}\right)$. Furthermore, even under a constant step-size and a fixed time horizon $T$, the term $\log(T/\delta)$ appears when $\theta > 1$. As noted earlier, this arises from the application of the sub-Weibull Freedman inequality, which may indicate that the $\log(T/\delta)$ dependence is intrinsic to problems with sub-Weibull-type gradient noise, rather than a byproduct of the analytical approach.

## 4.2 RIEMANNIAN CLIPPED-SsGD UNDER THE $p$-BCM NOISES

In this section, we give a high-probability convergence analysis of the clipped-SsGD method for weakly convex optimization on the Stiefel manifold under $p$-th moment type noise, as stated in Assumption 3.6. The reason for introducing the clipping technique is that gradient clipping is a core and effective method for dealing with heavy-tailed noise, and some existing works in the Euclidean space have shown that, under assumption 3.6, when $p \in (1, 2)$, the ordinary SsGD method may diverge.

Next, we consider the assumption that the gradient noise distribution possesses a bounded $p$-th central moment ($p$-BCM). Specifically, assuming the random vector $X$ is centered (i.e., $\mathbb{E}[X] = 0$), and for $p \in (1, 2]$, if

$$\mathbb{E}[\|X\|^p] \leq \sigma^p \leq +\infty, \quad (17)$$

we say $X$ has a bounded $p$-th central moment. The $p$-BCM noise distribution presents greater analytical challenges compared to the sub-Weibull noise assumption. Observe that when gradient noise satisfies the $p$-BCM condition with $p < 2$, its variance (i.e., the second central moment) may be unbounded. Thus, the $p$-BCM assumption is strictly weaker than the bounded-variance assumption. The case where $p \in (1, 2)$ is particularly challenging, as classical SGD theory indicates possible divergence in this case. Numerous prior studies in Euclidean settings—including Zhang et al. (2020), Cutkosky & Mehta (2021), Nguyen et al. (2023), and Gorbunov et al. (2023)—have shown that gradient clipping, which bounds the stochastic gradient norm by a threshold $\lambda_t$ at iteration $t$, enables convergence of certain stochastic first-order methods under $p$-BCM gradient noise in smooth or convex optimization. This approach is also extensively used in practical training. It is therefore natural to extend such analysis to weakly convex problems on Riemannian manifolds.

Now, we present the following framework of the Riemannian clipped-SsGD algorithm.

---

**Algorithm 2** Riemannian clipped-SsGD method

---

1: Input: $x_0 \in \text{St}(n, r)$, batch size $B > 0$, $\eta_0 > 0$, $\lambda_t > 0$.
2: **for** $t = 1, \cdots, T$ **do**
3:  Sample $\zeta_{1,t}, \cdots, \zeta_{B,t}$ independently from the distribution $D_\zeta$
4:  Access to unbiased Riemannian stochastic gradient oracle to get $\tilde{\nabla}_{\mathcal{R}} f(x_t, \zeta_{1,t}), \cdots,$ $\tilde{\nabla}_{\mathcal{R}} f(x_t, \zeta_{B,t})$ and compute $\overline{\tilde{\nabla}_{\mathcal{R}} f(x_t)} = \frac{1}{B} \sum_{i=1}^{B} \tilde{\nabla}_{\mathcal{R}} f(x_t, \zeta_{i,t})$.
5:  $\tilde{\nabla}_{\mathcal{R}} f(x_t) = \min \left\{ \frac{\lambda_t}{\|\overline{\tilde{\nabla}_{\mathcal{R}} f(x_t)}\|}, 1 \right\} \overline{\tilde{\nabla}_{\mathcal{R}} f(x_t)}, \xi_t = -\eta_t \tilde{\nabla}_{\mathcal{R}} f(x_t)$.
6:  $x_{t+1} = \mathcal{R}_{x_t}(\xi_t)$.
7: **end for**

---

Then we present a general high-probability convergence theorem of the Riemannian SsGD method under $p$-th moment type noise for weak convex optimization on the Stiefel manifold.

**Theorem 4.2.** *Suppose that Assumptions 3.1–3.4 and 3.6 hold. For any $\lambda, \eta_0 \in \mathbb{R}_+$, define $\lambda_t = \max\{2G, \lambda t^{1/p}\}$, $\quad \eta_t = \eta_0 \min\left\{\frac{1}{\lambda_t}, \frac{1}{G\sqrt{t}}\right\}$. Then, for $\bar{\rho} = 2(\rho+L)$ (where $L$ denotes the Lipschitz constant) and any batch size $B \in \mathbb{N} \setminus \{0\}$, the following inequality holds with probability at least $1 - \delta$:*

$$\frac{1}{T} \sum_{t=1}^{T} \|\Theta(x_t)\|^2 = O\left(\left\{\frac{\Delta_1}{\eta_0} + (G + \rho\eta_0)\left((\sigma/\lambda)^p B^{1-p}\log(T) + \log(1/\delta)\right)\right\}\right.$$
$$\left. \cdot \max\left\{\frac{\lambda}{T^{\frac{p-1}{p}}}, \frac{G}{\sqrt{T}}\right\}\right) \tag{18}$$

*If $\sigma$ is known, set $\lambda = \sigma$ and $B \equiv 1$, then we have*

$$\frac{1}{T} \sum_{t=1}^{T} \|\Theta(x_t)\|^2 = O\left(\left\{\frac{\Delta_1}{\eta_0} + (G + \rho\eta_0)\log(T/\delta)\right\} \cdot \max\left\{\frac{\lambda}{T^{\frac{p-1}{p}}}, \frac{G}{\sqrt{T}}\right\}\right) \tag{19}$$

*where $\Delta_1 \triangleq f_{1/2(\rho+L)}(x_1) - \min_{x \in \text{St}(n,r)} f(x)$.*

The results indicate that the high-probability convergence rate is $O(T^{\frac{1-p}{p}})$ with respect to $T$, independent of whether $\sigma$ is known. In the deterministic case where $\sigma = 0$, the step size becomes $\eta_t = \eta_0 \min\{1/G\sqrt{t}, 1/(2G)\} = O(1/\sqrt{t})$, and the clipping level reduces to $2G$. According to Assumption 3.3, $\lambda_t$ consistently bounds the norm of the sub-gradients of $f$ at all $x$, which implies that Algorithm 2 simplifies to the standard Riemannian gradient descent method, achieving a convergence rate of $O(1/\sqrt{T})$ in $T$. It is important to note that the $\log T$ term arises exclusively due to the time-varying nature of the step size. In contrast to Corollary 4.2, this logarithmic factor can be removed by pre-specifying a fixed time horizon $T$.

**Theorem 4.3.** *Suppose that Assumptions 3.1-3.4 and 3.6 hold, and fix the time horizon $T$. For any $\lambda, \eta_0 \in \mathbb{R}_+$, define $\lambda_t \equiv \max\{2G, \lambda T^{\frac{1}{p}}\}$ and $\eta_t = \eta_0 \min\left\{\frac{1}{\lambda_t}, \frac{1}{G\sqrt{T}}\right\}$. Then, for $\bar{\rho} = 2(\rho + L)$ (where $L$ denotes the Lipschitz constant) and any batch-size $B \in \mathbb{N} \setminus \{0\}$, the following inequality holds with the probability at least $1 - \delta$:*

$$\frac{1}{T} \sum_{t=1}^{T} \|\Theta(x_t)\|^2 = O\left(\left\{\frac{\Delta_1}{\eta_0} + (G + \rho\eta_0)\left((\sigma/\lambda)^p B^{1-p} + \log(1/\delta)\right)\right\}\right.$$
$$\left. \cdot \max\left\{\frac{\lambda}{T^{\frac{p-1}{p}}}, \frac{G}{\sqrt{T}}\right\}\right) \tag{20}$$

*If $\sigma$ is known, set $\lambda = \sigma$ and $B \equiv 1$, then we have*

$$\frac{1}{T} \sum_{t=1}^{T} \|\Theta(x_t)\|^2 = O\left(\left\{\frac{\Delta_1}{\eta_0} + (G + \rho\eta_0)\log(1/\delta)\right\} \cdot \max\left\{\frac{\sigma}{T^{\frac{p-1}{p}}}, \frac{G}{\sqrt{T}}\right\}\right) \tag{21}$$

*where $\Delta_1 \triangleq f_{1/2(\rho+L)}(x_1) - \min_{x \in \text{St}(n,r)} f(x)$.*

It is worth noting that, for a fixed time horizon $T$, the dependence on $\delta$ remains $\log(1/\delta)$, Regardless of whether $\sigma$ is known. This dependence is consistent with the high-probability convergence established in Zhu et al. (2025) for the clipped-SsGD method applied to weakly convex optimization in Euclidean space under $p$-th moment type noise.

Based on Theorems 4.2 and 4.3, we also provide an in-expectation convergence theorem in the following.

**Theorem 4.4.** *Suppose that Assumptions 3.1-3.4 and 3.6 hold. For any $\lambda, \eta_0 \in \mathbb{R}_+$, define $\lambda_t = \max\{2G, \lambda t^{\frac{1}{p}}\}$ and $\eta_t = \eta_0 \min\left\{\frac{1}{\lambda_t}, \frac{1}{G\sqrt{t}}\right\}$. Then, for $\bar{\rho} = 2(\rho+L)$ (where $L$ denotes the Lipschitz constant) and any batch-size $B \in \mathbb{N} \setminus \{0\}$, we have the following inequality*

$$\frac{1}{T}\sum_{t=1}^{T}\mathbb{E}_t\left[\|\Theta(x_t)\|^2\right] = O\left(\left\{\frac{\Delta_1}{\eta_0} + (\rho\eta_0 + G)(\sigma/\lambda)^p B^{1-p}\log(eT)\right.\right.$$

$$\left.\left.+\rho\eta_0\log(eT)\right\} \cdot \max\left\{\frac{\lambda}{T^{\frac{p-1}{p}}}, \frac{G}{\sqrt{T}}\right\}\right) \tag{22}$$

*If $\sigma$ is known, set $\lambda = \sigma$ and $B = 1$, then we get*

$$\frac{1}{T}\sum_{t=1}^{T}\mathbb{E}_t\left[\|\Theta(x_t)\|^2\right] = O\left(\left\{\frac{\Delta_1}{\eta_0} + (\rho\eta_0 + G)\log(eT)\right\} \cdot \max\left\{\frac{\sigma}{T^{\frac{p-1}{p}}}, \frac{G}{\sqrt{T}}\right\}\right) \tag{23}$$

Similar to Corollary 4.2, we can still provide the in-expectation convergence theorem after a fixed time-horizon $T$.

**Theorem 4.5.** *Suppose that Assumptions 3.1-3.4 and 3.6 hold. Fix the time-horizon $T$. For any $\lambda, \eta_0 \in \mathbb{R}_+$, define $\lambda_t \equiv \max\{2G, \lambda T^{\frac{1}{p}}\}$ and $\eta_t \equiv \eta_0 \min\left\{\frac{1}{\lambda_t}, \frac{1}{G\sqrt{T}}\right\}$. Then, for $\bar{\rho} = 2(\rho + L)$ (where $L$ denotes the Lipschitz constant) and any batch-size $B \in \mathbb{N} \setminus \{0\}$, we have the following inequality*

$$\frac{1}{T}\sum_{t=1}^{T}\mathbb{E}_t\left[\|\Theta(x_t)\|^2\right] = O\left(\left\{\frac{\Delta_1}{\eta_0} + (\rho\eta_0 + G)(\sigma/\lambda)^p B^{1-p}\right.\right.$$

$$\left.\left.+\rho\eta_0\right\} \cdot \max\left\{\frac{\lambda}{T^{\frac{p-1}{p}}}, \frac{G}{\sqrt{T}}\right\}\right) \tag{24}$$

*If $\sigma$ is known, set $\lambda = \sigma$ and $B = 1$, then we get*

$$\frac{1}{T}\sum_{t=1}^{T}\mathbb{E}_t\left[\|\Theta(x_t)\|^2\right] = O\left(\left\{\frac{\Delta_1}{\eta_0} + \rho\eta_0 + G\right\} \cdot \max\left\{\frac{\sigma}{T^{\frac{p-1}{p}}}, \frac{G}{\sqrt{T}}\right\}\right) \tag{25}$$

Theorem 4.5 implies that, in order to obtain an iterate point $x$ such that $\mathbb{E}\left[\|\Theta(x)\|\right] \leq \varepsilon$, the total number of subgradient evaluations needed is at most $O(\varepsilon^{-2p/(p-1)})$. Our result matches the known results for weakly convex optimization in Euclidean spaces. However, to the best of our knowledge, a lower bound on the sample complexity of stochastic subgradient methods under Assumption 3.6 has not been established for nonsmooth, weakly convex optimization on Riemannian manifolds. Hence, we contend that our finding retains its value for this more general setting.

## 5 CONCLUSION

This paper mainly investigates the high-probability convergence of Riemannian stochastic weakly convex optimization under heavy-tailed subgradient noise. First, we analyze the high-probability convergence of the general Riemannian stochastic subgradient descent (SsGD) for weakly convex optimization on the Stiefel manifold under sub-Weibull-type subgradient noise. Then, under the assumption that the subgradient noise has a $p$-th bounded central moment ($p$-BCM), where $p \in (1, 2]$, we establish the high-probability and in-expectation convergence of Riemannian clipped-SsGD. Our research results provide theoretical guarantees for these methods in nonsmooth weakly convex optimization on Riemannian manifolds under weaker noise assumptions.

ETHICS STATEMENT

This research is a theoretical study of Riemannian stochastic weakly convex optimization under heavy-tailed noises. It does not involve any human subjects, private data, or experiments with obvious societal risks. We see no direct ethical concerns arising from this work itself.

REPRODUCIBILITY STATEMENT

(i) We provide a proof sketch for Theorem 4.1 in App. B, and present the detailed proof process for Theorem 4.1 in App. C.1. (ii) We provide a proof sketch for Theorem 4.2 in App. D, and present the detailed proof process for Theorem 4.2 in App. E.1.

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

# A CONCENTRATION INEQUALITIES

Here we present the concentration inequality in Madden et al. (2024) to assist us in analyzing the high-probability convergence of Algorithm 1 and Algorithm 2.

**Theorem A.1.** *Madden et al. (2024) Consider a filtered probability space $(\Omega, \mathcal{F}, (\mathcal{F}_t), P)$. Let $\{X_t\}$ and let the processes $\{\sigma_t\}$ be adapted to $\mathcal{F}_t \subset \mathcal{F}$ such that for any $t \in [T]$, $\mathbb{E}[X_t|\mathcal{F}_{t-1}] = 0$ and $\sigma_t \geq 0$ almost surely. Suppose $X_t$ is $\sigma_{t-1}$-sub-Weibull($\theta$) with parameter $\theta \geq 1/2$, meaning*

$$\mathbb{E}\left[\exp\left\{(|X_t|/\sigma_{t-1})^{1/\theta}\right\} \mid \mathcal{F}_{t-1}\right] \leq 2,$$

*and assume $\sigma_{t-1} \leq M_t$ holds almost surely whenever $\theta > 1/2$. Then, for all $x, \beta \geq 0$ and $\varepsilon \in (0, 1)$, the following inequality holds:*

$$P\left(\bigcup_{\tau \in [T]} \left\{\sum_{t=1}^{\tau} X_t \geq x \text{ and } \sum_{t=1}^{\tau} a\sigma_{t-1}^2 \leq \alpha \sum_{t=1}^{\tau} X_t + \beta\right\}\right) \leq \exp(-\lambda x + 2\lambda^2 \beta) + 2\varepsilon,$$

*provided that $\alpha \geq b \max_{t \in [T]} M_t$ and $\lambda \in [0, \frac{1}{2\alpha}]$, where $a$ and $b$ are defined as follows:*

$$a = \begin{cases} 2, & \theta = \frac{1}{2} \\ (4\theta)^{2\theta} e^2, & \theta \in (\frac{1}{2}, 1] \\ (2^{2\theta+1} + 2)\Gamma(2\theta + 1) + \frac{2^{3\theta}\Gamma(3\theta+1)}{3\log(T/\varepsilon)^{\theta-1}}, & \theta > 1 \end{cases}$$

$$b = \begin{cases} 0, & \theta = \frac{1}{2} \\ (4\theta)^{\theta}, & \theta \in (\frac{1}{2}, 1] \\ 2\log(T/\varepsilon)^{\theta-1}, & \theta > 1 \end{cases}$$

**Lemma A.1.** *Vladimirova et al. (2020) Consider a sequence of random variables $X_1, X_2, \ldots, X_T$, where for each $t \in [T]$, $X_t$ is $\sigma_t$-sub-Weibull($\theta$) ($\theta \geq 1/2$). Then, for all $\gamma \geq 0$, we get*

$$P\left(\left|\sum_{t=1}^{T} X_t\right| \geq \gamma\right) \leq 2\exp\left\{-\left(\frac{\gamma}{v(\theta)\sum_{t=1}^{T}\sigma_t}\right)^{1/\theta}\right\}, \tag{26}$$

*where*

$$v(\theta) = \begin{cases} (4e)^{\theta}, & \theta \leq 1 \\ 2(2e\theta)^{\theta}, & \theta \geq 1 \end{cases} \tag{27}$$

**Corollary A.1.** *Suppose that $X_1, X_2, \ldots, X_T$ satisfy the conditions in Lemma A.1. Then, with the probability at least $1 - \delta$, we have following inequality*

$$\left|\sum_{t=1}^{T} X_t\right| \leq v(\theta) \sum_{t=1}^{T} \sigma_t \log\left(\frac{2}{\delta}\right)^{\theta}, \tag{28}$$

*where $v(\theta)$ is as defined in Lemma A.1.*

This corollary follows directly from setting $\gamma = v(\theta) \sum_{t=1}^{T} \sigma_t \log(\frac{2}{\delta})^{\theta}$ in Lemma A.1. It is worth emphasizing that neither Lemma A.1 nor Corollary A.1 assumes $X_t$ to be a martingale.

**Lemma A.2.** *Freedman (1975) Consider a filtered probability space $(\Omega, \mathcal{F}, (\mathcal{F}_t), P)$, and let $\{X_t\}_{t=1}^{T}$ be a martingale difference sequence adapted to $(\mathcal{F}_t)$. Suppose that for all $t \in \mathbb{N}_+$, $|X_t| \leq R$ almost surely, and for some fixed $T \in \mathbb{N}_+$, $\sum_{t=1}^{T} \mathbb{E}[|X_t|^2|\mathcal{F}_{t-1}] \leq F$ holds with probability 1. Then, for any $\tau \in [T]$, we have that the following inequality holds with probability at least $1 - \delta$:*

$$\sum_{t=1}^{T} |X_t| \leq \frac{2R\log(2/\delta)}{3} + \sqrt{2F\log(2/\delta)}. \tag{29}$$

## B  PROOF SKETCH OF THEOREM 4.1

Here we briefly summarize convergence proof of the Riemannian SsGD method for weakly convex optimization under sub-Weibull-type noise. To facilitate the analysis of the stochastic process, we define $\mathcal{F}_t = \sigma(g_1, \ldots, g_t)$, with $\mathcal{F}_t$ residing on the probability space $(\Omega, \mathcal{F}, P)$. Here, $\mathbb{E}_t[\cdot]$ represents the conditional expectation with respect to $\mathcal{F}_{t-1}$. First, we present the Riemannian subgradient inequality, which is applicable to the restrictions of weakly convex functions on the Stiefel manifold.

**Lemma B.1.** *Li et al. (2021) For any bounded open set $\mathcal{U}$ that contains $\mathrm{St}(n, r)$, there exists a constant $L > 0$ such that $f$ is $L$-Lipschitz continuous on $\mathcal{U}$, and following inequality holds for all $x, y \in \mathrm{St}(n, r)$ and any Riemannian subgradient $\tilde{\nabla}_\mathcal{R} f(x) \in \partial_\mathcal{R} f(x)$:*

$$f(y) \geq f(x) + \langle \tilde{\nabla}_\mathcal{R} f(x), y - x \rangle - \frac{\rho + L}{2} \|y - x\|^2. \tag{30}$$

Next, we present a basic lemma for a SsGD method under Riemannian weak convex optimization.

**Lemma B.2.** *Suppose that Assumption 3.1, 3.2, 3.3 and 3.4 hold. For any $t \in [T]$ and $\bar{\rho} > \rho$, the sequence $x_t$ generated by the Riemannian SsGD method satisfies the following inequality:*

$$\frac{\bar{\rho} - \rho - L}{\bar{\rho}} \eta_t \|\Theta(x_t)\|^2 \leq \Delta_t - \Delta_{t+1} + \bar{\rho} \eta_t \langle \hat{x}_t - x_t, \xi_t \rangle + \bar{\rho} \eta_t^2 (\|\xi_t\|^2 + G^2) \tag{31}$$

*where $\Delta_t \triangleq f_{1/\bar{\rho}}(x_t) - \min_{x \in \mathrm{St}(n,r)} f(x), f(\hat{x}_t) = f_{1/\bar{\rho}}(x_t), \partial_t \triangleq \mathbb{E}_t\left[\tilde{\nabla}_\mathcal{R} f(x)\right] \in \partial_\mathcal{R} f(x_t)$ and $\xi_t \triangleq \tilde{\nabla}_\mathcal{R} f(x) - \partial_t$.*

*Proof.* By the definition of $\hat{x}_{t+1}$, we have

$$f_{1/\bar{\rho}}(x_{t+1}) = f(\hat{x}_{t+1}) + \frac{\bar{\rho}}{2} \|\hat{x}_{t+1} - x_{t+1}\|^2$$

$$\leq f(\hat{x}_t) + \frac{\bar{\rho}}{2} \|\hat{x}_t - x_{t+1}\|^2$$

$$= f(\hat{x}_t) + \frac{\bar{\rho}}{2} \left\|\hat{x}_t - \mathcal{R}_{x_t}\left(-\eta_t \tilde{\nabla}_\mathcal{R} f(x_t)\right)\right\|^2$$

$$\leq f(\hat{x}_t) + \frac{\bar{\rho}}{2} \left\|x_t - \eta_t \tilde{\nabla}_\mathcal{R} f(x_t) - \hat{x}_t\right\|^2$$

$$= f(\hat{x}_t) + \frac{\bar{\rho}}{2} \|\hat{x}_t - x_t\|^2 + \bar{\rho} \eta_t \left\langle \hat{x}_t - x_t, \tilde{\nabla}_\mathcal{R} f(x_t) \right\rangle + \frac{\bar{\rho} \eta_t^2}{2} \left\|\tilde{\nabla}_\mathcal{R} f(x_t)\right\|^2$$

$$= f_{1/\bar{\rho}}(\hat{x}_t) + \bar{\rho} \eta_t \langle \hat{x}_t - x_t, \partial_t \rangle + \bar{\rho} \eta_t \langle \hat{x}_t - x_t, \xi_t \rangle + \frac{\bar{\rho} \eta_t^2}{2} \|\partial_t + \xi_t\|^2$$

$$\leq f_{1/\bar{\rho}}(\hat{x}_t) + \bar{\rho} \eta_t \left(f(\hat{x}_t) - f(x_t) + \frac{\rho + L}{2} \|\hat{x}_t - x_t\|^2\right) + \bar{\rho} \eta_t \langle \hat{x}_t - x_t, \xi_t \rangle + \bar{\rho} \eta_t^2 \left(\|\xi_t\|^2 + G^2\right)$$

$$= f_{1/\bar{\rho}}(\hat{x}_t) - \bar{\rho} \eta_t \left[\left(f(x_t) + \frac{\bar{\rho}}{2} \|x_t - x_t\|^2\right) - \left(f(\hat{x}_t) + \frac{\bar{\rho}}{2} \|x_t - \hat{x}_t\|^2\right) + \frac{\bar{\rho} - \rho - L}{2} \|x_t - \hat{x}_t\|^2\right]$$

$$+ \bar{\rho} \eta_t \langle \hat{x}_t - x_t, \xi_t \rangle + \bar{\rho} \eta_t^2 \left(\|\xi_t\|^2 + G^2\right)$$

$$\leq f_{1/\bar{\rho}}(\hat{x}_t) - \bar{\rho} \eta_t (\bar{\rho} - \rho - L) \|\hat{x}_t - x_t\|^2 + \bar{\rho} \eta_t \langle \hat{x}_t - x_t, \xi_t \rangle + \bar{\rho} \eta_t^2 \left(\|\xi_t\|^2 + G^2\right)$$

$$= f_{1/\bar{\rho}}(\hat{x}_t) - \frac{\bar{\rho} - \rho - L}{\bar{\rho}} \eta_t \|\Theta(x_t)\|^2 + \bar{\rho} \eta_t \langle \hat{x}_t - x_t, \xi_t \rangle + \bar{\rho} \eta_t^2 \left(\|\xi_t\|^2 + G^2\right)$$

where the second inequality holds because properties of polar retraction in (Li et al., 2021, Lemma 1), the third inequality follow the Riemannian subgradient inequality of $f$ (see Lemma B.1), the fourth inequality is due to $\triangleq f(x) + \frac{\bar{\rho}}{2} \|x - x_t\|^2$ is a $\frac{\bar{\rho} - \rho - L}{2}$-strongly convex function. By the definition of strongly convexity, we have

$$\left(f(x_t) + \frac{\bar{\rho}}{2} \|x_t - x_t\|^2\right) - \left(f(\hat{x}_t) + \frac{\bar{\rho}}{2} \|x_t - \hat{x}_t\|^2\right)$$

$$\geq \left\langle \hat{\partial}_t + \bar{\rho}(\hat{x}_t - x_t), x_t - \hat{x}_t \right\rangle + \frac{\bar{\rho} - \rho - L}{2} \|x_t - \hat{x}_t\|^2$$

where $\hat{\partial}_t \in \partial f(\hat{x}_t)$. Since $\hat{x}_t$ minimizes the function $f(x) + \frac{\bar{\rho}}{2}|x - x_t|^2$ on $\text{St}(n, r)$, the optimality condition ensures that $\left\langle \hat{\partial}_t + \bar{\rho}(\hat{x}_t - x_t), x_t - \hat{x}_t \right\rangle$ is non-negative. The last inequality holds because we use (5). Rearranging the terms yields the desired result, and thus we complete the proof. □

An analogous convergence result to (31) has been established in several studies that analyze the in-expectation convergence of the SsGD method for weakly convex optimization in Euclidean space, including Davis & Drusvyatskiy (2018; 2019); Alacaoglu et al. (2020); Hu et al. (2023). In contrast to in-expectation convergence analysis, the martingale term $\bar{\rho}\eta_t\langle \hat{x}_t - x_t, \xi_t \rangle$ (i.e., $\mathbb{E}_t[\bar{\rho}\eta_t\langle \hat{x}_t - x_t, \xi_t \rangle] = 0$) and its probability upper bound must be established via concentration inequalities. Therefore, we present the following lemma, which can be directly derived from the concentration inequality in Madden et al. (2024).

**Lemma B.3.** *Consider a filtered probability space $(\Omega, \mathcal{F}, (\mathcal{F}_t), P)$. Let the processes $\{X_t\}$ and $\{\sigma_t\}$ be adapted to $\mathcal{F}_t \subset \mathcal{F}$, such that for any $t \in [T]$, $\mathbb{E}[X_t \mid \mathcal{F}_{t-1}] = 0$ and $\sigma_t$ is almost surely nonnegative. Suppose that $X_t$ is $\sigma_{t-1}$-sub-Weibull($\theta$) with parameter $\theta \geq \frac{1}{2}$ and that $\sigma_{t-1} \leq M_t$ almost surely when $\theta > \frac{1}{2}$. Then, for any $\alpha \geq b \max_{t \in [T]} M_t$, the following inequality holds with the probability at least $1 - \delta$.*

$$\sum_{t=1}^{T} X_t \leq 2\alpha \log\left(\frac{2}{\delta}\right) + \frac{a}{\alpha}\sum_{t=1}^{T}\sigma_{t-1}^2, \tag{32}$$

*where $a$ and $b$ are defined as following:*

$$a = \begin{cases} 2, & \theta = \frac{1}{2} \\ (4\theta)^{2\theta}e^2, & \theta \in \left(\frac{1}{2}, 1\right] \\ \left(2^{2\theta+1} + 2\right)\Gamma(2\theta+1) + \frac{2^{3\theta}\Gamma(3\theta+1)}{3\log(4T/\delta)^{\theta-1}}, & \theta > 1 \end{cases}$$

$$b = \begin{cases} 0, & \theta = \frac{1}{2} \\ (4\theta)^\theta, & \theta \in \left(\frac{1}{2}, 1\right] \\ 2\log(4T/\delta)^{\theta-1}, & \theta > 1 \end{cases}$$

Note that the term $\bar{\rho}\eta_t\langle \hat{x}_t - x_t, \xi_t \rangle$ is $\bar{\rho}\eta_t\|\hat{x}_t - x_t\|\sigma$-sub-Weibull($\theta$). In order to use Lemma B.2 for $\bar{\rho}\eta_t\langle \hat{x}_t - x_t, \xi_t \rangle$, it is also necessary that $\bar{\rho}\eta_t\|\hat{x}_t - x_t\|\sigma$ be bounded above by $M_t$ in the case where $\theta > 1/2$. Thus, we give the following lemma.

**Lemma B.4.** *Given a $\rho$-weakly convex function $f$, for any $\bar{\rho} > \rho$, let $f_{1/\bar{\rho}}(x)$ denote the Moreau envelope of $f(x)$. Then, for every $\tilde{\nabla}_{\mathcal{R}}f(x) \in \partial_{\mathcal{R}}f(x)$, the following inequality holds*

$$\|\hat{x} - x\| \leq \frac{2\|\tilde{\nabla}_{\mathcal{R}}f(x)\|}{\bar{\rho} - \rho - L}. \tag{33}$$

*Proof.* Since $\hat{x} = \arg\min_{y \in \text{St}(n,r)}\left\{f(y) + \frac{\bar{\rho}}{2}\|y - x\|^2\right\}$

$$f(\hat{x}) + \frac{\bar{\rho}}{2}\|\hat{x} - x\|^2 \leq f(x) + \frac{\bar{\rho}}{2}\|x - x\|^2 \tag{34}$$

For any $\tilde{\nabla}_{\mathcal{R}}f(x) \in \partial_{\mathcal{R}}f(x)$, according to the definition of the $\rho$-weakly convexity, we get

$$f(\hat{x}) - f(x) \geq \langle \tilde{\nabla}_{\mathcal{R}}f(x), \hat{x} - x \rangle - \frac{\rho + L}{2}\|\hat{x} - x\|^2$$

Combining these two inequalities, we have

$$\frac{\bar{\rho} - \rho - L}{2}\|\hat{x} - x\|^2 \leq \langle \tilde{\nabla}_{\mathcal{R}}f(x), x - \hat{x} \rangle \leq \|\tilde{\nabla}_{\mathcal{R}}f(x)\|\|x - \hat{x}\|$$

$$\Longrightarrow \|\hat{x} - x\| \leq \frac{2\|\tilde{\nabla}_{\mathcal{R}}f(x)\|}{\bar{\rho} - \rho - L}$$

□

By combining Lemma B.4 and Assumption 3.3, we can establish an upper bound for $\bar{\rho}\eta_t\|\hat{x}_t - x_t\|\sigma$. This allows us to apply the concentration inequality (32) to bound the term $\bar{\rho}\eta_t\langle x_{t+1} - x_t, \xi_t \rangle$ by $O(\eta_t^2\|\Theta(x_t)\|^2)$, which differs from the left side of (31) in that it is a decreasing term.

## C  RIEMANNIAN SSGD UNDER THE $\sigma$-SUB-WEIBULL($\theta$) NOISES

### C.1  PROOF OF THEOREM 4.1

*Proof.* We begin by summing both sides of equation (31) in Lemma B.2 from $t = 1$ to $T$.

$$\frac{\bar{\rho} - \rho - L}{\bar{\rho}} \sum_{t=1}^{T} \eta_t \|\Theta(x_t)\|^2 \leq \Delta_1 + \sum_{t=1}^{T} \bar{\rho} \eta_t \langle \hat{x}_t - x_t, \xi_t \rangle + \sum_{t=1}^{T} \bar{\rho} \eta_t^2 \|\xi_t\|^2 + \bar{\rho} G^2 \sum_{t=1}^{T} \eta_t^2 \quad (35)$$

Note that since $\forall x \in \text{St}(n,r)$, $f_{1/\bar{\rho}}(x) \geq \min_{x \in \text{St}(n,r)} f$, we get $\Delta_1 - \Delta_{T+1} \leq \Delta_1$.

First of all, we analyze $\sum_{t=1}^{T} \bar{\rho} \eta_t^2 \|\xi_t\|^2$. According to the Assumption 3.5, it follows that

$$\mathbb{E}\left[\exp\left\{\left(\frac{\bar{\rho} \eta_t^2 \|\xi_t\|^2}{\bar{\rho} \eta_t^2 \sigma^2}\right)^{\frac{1}{2\theta}}\right\}\right] = \mathbb{E}\left[\mathbb{E}_t\left[\exp\left\{\left(\frac{\bar{\rho} \eta_t^2 \|\xi_t\|^2}{\bar{\rho} \eta_t^2 \sigma^2}\right)^{\frac{1}{2\theta}}\right\}\right]\right] \leq 2 \quad (36)$$

which implies that $\bar{\rho} \eta_t^2 \|\xi_t\|^2$ is $\eta_t^2 \sigma^2 \bar{\rho}$-sub-Weibull($2\theta$). Applying Lemma A.1, we obtain the following inequality with probability at least 1-$\delta_1$

$$\sum_{t=1}^{T} \bar{\rho} \eta_t^2 \|\xi_t\|^2 \leq v(2\theta) \sum_{t=1}^{T} \eta_t^2 \sigma^2 \bar{\rho} \log\left(\frac{2}{\delta_1}\right)^{2\theta} \quad (37)$$

Observe that since $\theta \geq 1/2$, we have $v(2\theta) = 2(4e\theta)^{2\theta}$. This implies that

$$\sum_{t=1}^{T} \bar{\rho} \eta_t^2 \|\xi_t\|^2 \leq 2\bar{\rho}\sigma^2 (4e\theta \log 2/\delta_1)^{2\theta} \sum_{t=1}^{T} \eta_t^2 \quad (38)$$

We now derive a probability upper bound for the term $\sum_{t=1}^{T} \bar{\rho} \eta_t \langle \hat{x}_t - x_t, \xi_t \rangle$. Observe that $\mathbb{E}_t[\bar{\rho} \eta_t \langle \hat{x}_t - x_t, \xi_t \rangle] = 0$, indicating that $\bar{\rho} \eta_t \langle \hat{x}_t - x_t, \xi_t \rangle$ forms a martingale difference sequence. Let

$$\sigma_{t-1} = \eta_t \bar{\rho} \|\hat{x}_t - x_t\| \sigma$$
$$M_t = \frac{2\bar{\rho} G \sigma \eta_t}{\bar{\rho} - \rho - L}$$

Note that applying the Cauchy–Schwarz inequality yields $\bar{\rho} \eta_t \langle \hat{x}_t - x_t, \xi_t \rangle \leq \sigma_{t-1}$ almost surely, where $\sigma_t$ is adapted to $\mathcal{F}_t$. By Lemma B.4 and Assumption 3.3, it follows that $\sigma_{t-1} \leq M_t$ for all $t \in [T]$. and we obtain

$$\mathbb{E}_t\left[\exp\left\{(\bar{\rho} \eta_t |\langle \hat{x}_t - x_t, \xi_t \rangle|/\sigma_{t-1})^{1/\theta}\right\}\right] \leq \mathbb{E}_t\left[\exp\left\{(\|\xi_t\|/\sigma)^{1/\theta}\right\}\right] \leq 2 \quad (39)$$

which means $\bar{\rho} \eta_t \langle \hat{x}_t - x_t, \xi_t \rangle$ is $\sigma_{t-1}$-sub-Weibull($\theta$). Applying Lemma B.3, we obtain the following inequality with probability at least $1 - \delta_2$

$$\sum_{t=1}^{T} \bar{\rho} \eta_t \langle \hat{x}_t - x_t, \xi_t \rangle \leq 2\alpha \log \frac{2}{\delta_2} + \frac{a}{\alpha} \sum_{t=1}^{T} \sigma^2 \eta_t^2 \bar{\rho}^2 \|\hat{x}_t - x_t\|^2$$

$$= 2\alpha \log \frac{2}{\delta_2} + \frac{a}{\alpha} \sum_{t=1}^{T} \sigma^2 \eta_t^2 \|\Theta(x_t)\|^2 \quad (40)$$

By substituting (38) and (40) into (35), we obtain the following inequality, which holds with probability at least $1 - \delta_1 - \delta_2$

$$\sum_{t=1}^{T} \eta_t \left(\frac{\bar{\rho} - \rho - L}{\bar{\rho}} - \frac{a\sigma^2}{\alpha} \eta_t\right) \|\Theta(x_t)\|^2 \leq \Delta_1 + 2\alpha \log(2/\delta_2)$$

$$+ \left(2\bar{\rho}\sigma^2 (4e\theta \log(2/\delta_1))^{2\theta} + \bar{\rho} G^2\right) \sum_{t=1}^{T} \eta_t^2$$

$$(41)$$

Given that both $a$ and $\alpha$ depend on the value of $\theta$, we analyze the probability upper bounds for different ranges of $\theta$. For simplicity, we take $\bar{\rho} = 3(\rho + L)$.

**Case 1.** $\theta = \frac{1}{2}$

For $\theta = 1/2$, we have $a = 2, b = 0$ and $\alpha$ is only required to be nonnegative. Setting $\alpha = 6\sigma^2 \max_{t \in [T]} \eta_t$, it follows that

$$\frac{\bar{\rho} - \rho - L}{\bar{\rho}} - \frac{a\sigma^2}{\alpha}\eta_t = \frac{2}{3} - \frac{\eta_t}{3\max_{t \in [T]} \eta_t} \geq \frac{1}{3} \tag{42}$$

Then we get

$$\frac{1}{3}\sum_{t=1}^{T} \eta_t \|\Theta(x_t)\|^2 \leq \Delta_1 + 12\sigma^2 \max_{t \in [T]} \eta_t \log(2/\delta_2)$$

$$+ \left(12e\log(2/\delta_1)\sigma^2 + 3G^2\right)\sum_{t=1}^{T}(\rho + L)\eta_t^2 \tag{43}$$

Setting $\delta_1 = \delta_2 = \delta/2$ and multiplying both sides by $3\left(\sum_{t=1}^{T} \eta_t\right)^{-1}$, we obtain

$$\sum_{t=1}^{T} \frac{\eta_t}{\sum_{t=1}^{T} \eta_t}\|\Theta(x_t)\|^2 \leq \frac{3\Delta_1 + 36\sigma^2 \max_{t \in [T]} \eta_t \log(4/\delta)}{\sum_{t=1}^{T} \eta_t}$$

$$+ \left(36e\log(4/\delta)\sigma^2 + 9G^2\right)\frac{\sum_{t=1}^{T}(\rho + L)\eta_t^2}{\sum_{t=1}^{T} \eta_t} \tag{44}$$

**Case 2.** $\theta \in (\frac{1}{2}, 1]$

For $\theta \in (1/2, 1]$, it holds that $a = (4\theta)^{2\theta}e^2$ and $\alpha$ must satisfy $(4\theta)^{\theta} \max_{t \in [T]} M_t = 2(4\theta)^{\theta}\bar{\rho}G\sigma \max_{t \in [T]} \eta_t/(\bar{\rho} - \rho - L) = 3(4\theta)^{\theta}G\sigma \max_{t \in [T]} \eta_t$.

Setting $\alpha = \max\{3(4\theta)^{2\theta}e^2\sigma^2, 3(4\theta)^{\theta}G\sigma\} \max_{t \in [T]} \eta_t$, then we get

$$\frac{\bar{\rho} - \rho - L}{\bar{\rho}} - \frac{a\sigma^2}{\alpha}\eta_t = \frac{2}{3} - \frac{(4\theta)^{2\theta}e^2\sigma^2}{3\max\{(4\theta)^{2\theta}e^2\sigma^2, (4\theta)^{\theta}G\sigma\}} \cdot \frac{\eta_t}{\max_{t \in [T]} \eta_t} \geq \frac{1}{3} \tag{45}$$

Thus, the following inequality holds with the probability at least $1 - \delta_1 - \delta_2$:

$$\frac{1}{3}\sum_{t=1}^{T} \eta_t \|\Theta(x_t)\|^2 \leq \Delta_1 + 6(4\theta)^{\theta} \max\{(4\theta)^{\theta}e^2\sigma^2, G\sigma\}\log(2/\delta_2)\max_{t \in [T]} \eta_t$$

$$+ \left(6\sigma^2(4e\theta\log(2/\delta_1))^{2\theta} + 3G^2\right)\sum_{t=1}^{T}(\rho + L)\eta_t^2$$

$$\leq \Delta_1 + 24\max\{4e^2\sigma^2, G\sigma\}\log(2/\delta_2)\max_{t \in [T]} \eta_t$$

$$+ \left(96e^2\sigma^2(\log(2/\delta_1))^{2\theta} + 3G^2\right)\sum_{t=1}^{T}(\rho + L)\eta_t^2$$

$$\leq \Delta_1 + \max\left\{710\sigma^2, 24G\sigma\right\}\log(2/\delta_2)\max_{t \in [T]} \eta_t$$

$$+ \left(710\sigma^2(\log(2/\delta_1))^{2\theta} + 3G^2\right)\sum_{t=1}^{T}(\rho + L)\eta_t^2 \tag{46}$$

where the second inequality follows from the fact that for $\theta \in (1/2, 1]$, the relation $\theta^{2\theta} \leq \theta^\theta \leq \sqrt{\theta}$ holds. Setting $\delta_1 = \delta_2 = \delta_3 = \delta/2$ and multiplying both sides by $3 \left( \sum_{t=1}^T \eta_t \right)^{-1}$, we get

$$
\sum_{t=1}^T \frac{\eta_t}{\sum_{t=1}^T \eta_t} \|\Theta(x_t)\|^2 \leq \frac{3\Delta_1 + \max\{2130\sigma^2, 72G\sigma\} \log(4/\delta) \max_{t \in [T]} \eta_t}{\sum_{t=1}^T \eta_t}
$$
$$
+ \left( 2130\sigma^2 (\log(4/\delta))^{2\theta} + 9G^2 \right) \frac{\sum_{t=1}^T (\rho + L)\eta_t^2}{\sum_{t=1}^T \eta_t} \tag{47}
$$

**Case 3.** $\theta > 1$

For $\theta > 1$, we have $a = (2^{2\theta+1} + 2)\Gamma(2\theta + 1) + \frac{2^{3\theta}\Gamma(3\theta+1)}{3 \log(4T/\delta_3)^{\theta-1}}$ and it is required that $\alpha \geq 6G\sigma \log(4T/\delta_3)^{\theta-1} \max_{t \in [T]} \eta_t$.

We define $\alpha = \max \left\{ 3(2^{2\theta+1} + 2)\Gamma(2\theta + 1)\sigma^2 + \frac{2^{3\theta}\Gamma(3\theta+1)\sigma^2}{\log(4T/\delta_3)^{\theta-1}}, 6G\sigma \log(4T/\delta_3)^{\theta-1} \right\}$ $\max_{t \in [T]} \eta_t$, ensuring that $\frac{\bar{\rho} - \rho - L}{\bar{\rho}} - \frac{a\sigma^2\eta_t}{\alpha} \geq 1/3$. Then, the following inequality holds with the probability at least $1 - \delta_1 - \delta_2$:

$$
\frac{1}{3} \sum_{t=1}^T \eta_t \|\Theta(x_t)\|^2 \leq \Delta_1 + \max \left\{ 6(2^{2\theta+1} + 2)\Gamma(2\theta + 1)\sigma^2 + \frac{2^{3\theta}\Gamma(3\theta + 1)\sigma^2}{\log(4T/\delta_2)^{\theta-1}}, \right.
$$
$$
\left. 12G\sigma \log(4T/\delta_2)^{\theta-1} \right\} \log(2/\delta_2) \max_{t \in [T]} \eta_t
$$
$$
+ \left( 6\sigma^2 (4e\theta \log(2/\delta_1))^{2\theta} + 3G^2 \right) \sum_{t=1}^T (\rho + L)\eta_t^2
$$
$$
\leq \Delta_1 + \max \left\{ 5 \times 2^{3\theta+1}\Gamma(3\theta + 1)\sigma^2, 12G\sigma \log(4T/\delta_2)^{\theta-1} \right\} \cdot \log(2/\delta_2) \max_{t \in [T]} \eta_t
$$
$$
+ \left( 6(11\theta \log(2/\delta_1))^{2\theta}\sigma^2 + 3G^2 \right) \sum_{t=1}^T (\rho + L)\eta_t^2 \tag{48}
$$

Setting $\delta_1 = \delta_2 = \delta/2$ and multiplying both sides by $3 \left( \sum_{t=1}^T \eta_t \right)^{-1}$, we get

$$
\sum_{t=1}^T \frac{\eta_t}{\sum_{t=1}^T \eta_t} \|\Theta(x_t)\|^2 \leq \frac{3\Delta_1 + \hat{D}(\theta) \log(4/\delta) \max_{t \in [T]} \eta_t}{\sum_{t=1}^T \eta_t}
$$
$$
+ \left( 18(11\theta \log(4/\delta))^{2\theta}\sigma^2 + 9G^2 \right) \frac{\sum_{t=1}^T (\rho + L)\eta_t^2}{\sum_{t=1}^T \eta_t} \tag{49}
$$

where $\hat{D}(\theta) = \max \left\{ 15 \times 2^{3\theta+1}\Gamma(3\theta + 1)\sigma^2, 36G\sigma \log(8T/\delta)^{\theta-1} \right\}$. $\qquad \square$

## C.2 PROOF OF COROLLARY 4.1

*Proof.* By setting $\eta_t = \frac{\gamma}{\sqrt{t}}$, we have $\max_{t \in [T]} \eta_t = \gamma$. The analysis proceeds similarly to Theorem 4.1, with the sole modification that both sides are multiplied by $3(\eta_T T)^{-1}$ in the final step for each case. Additionally, we observe that $\sum_{t=1}^T \eta_t^2 \leq \gamma^2 \log(eT)$. Consequently, the following results hold:

- For $\theta = \frac{1}{2}$, the following inequality holds:

$$
\frac{1}{T} \sum_{t=1}^T \|\Theta(x_t)\|^2 \leq \frac{3\Delta_1}{\gamma\sqrt{T}} + \frac{36 \log(4/\delta)\sigma^2}{\sqrt{T}}
$$
$$
+ \gamma(\rho + L) \left( 36e \log(4/\delta)\sigma^2 + 9G^2 \right) \frac{\log(eT)}{\sqrt{T}} \tag{50}
$$

- For $\theta \in (\frac{1}{2}, 1]$, the following inequality holds:

$$\frac{1}{T}\sum_{t=1}^{T}\|\Theta(x_t)\|^2 \leq \frac{3\Delta_1}{\gamma\sqrt{T}} + \frac{\max\{2130\sigma^2, 72G\sigma\}\log(4/\delta)}{\sqrt{T}}$$

$$+ \gamma(\rho + L)\left(2130(\log(4/\delta))^{2\theta}\sigma^2 + 9G^2\right)\frac{\log(eT)}{\sqrt{T}} \quad (51)$$

- For $\theta > 1$, the following inequality holds:

$$\frac{1}{T}\sum_{t=1}^{T}\|\Theta(x_t)\|^2 \leq \frac{3\Delta_1}{\gamma\sqrt{T}} + \frac{\hat{D}(\theta)\log(4/\delta)}{\sqrt{T}}$$

$$+ \gamma(\rho + L)\left(18(11\theta\log(4/\delta))^{2\theta}\sigma^2 + 9G^2\right)\frac{\log(eT)}{\sqrt{T}} \quad (52)$$

$\square$

## C.3 PROOF OF COROLLARY 4.2

*Proof.* If we fix $\eta_t$ as $\eta$, then $\max_{t\in[T]} \eta_t = \eta$. In this case, Theorem 4.1 reduces to the following form:

- For $\theta = \frac{1}{2}$, we get

$$\frac{1}{T}\sum_{t=1}^{T}\|\Theta(x_t)\|^2 \leq \frac{3\Delta_1}{T\eta} + \frac{36\sigma^2\log(4/\delta)}{T} + (\rho + L)\left(36e\log(4/\delta)\sigma^2 + 9G^2\right)\eta \quad (53)$$

Then we set $\eta = \sqrt{\frac{3\Delta_1}{(\rho+L)(36e\log(4/\delta)\sigma^2+9G^2)T}}$, and we have

$$\frac{1}{T}\sum_{t=1}^{T}\|\Theta(x_t)\|^2 \leq 2\sqrt{\frac{(\rho + L)\Delta_1\left(108e\log(4/\delta)\sigma^2 + 27G^2\right)}{T}} + \frac{36\log(4/\delta)\sigma^2}{T} \quad (54)$$

- For $\theta \in (\frac{1}{2}, 1]$, we get

$$\frac{1}{T}\sum_{t=1}^{T}\|\Theta(x_t)\|^2 \leq \frac{3\Delta_1}{T\eta} + \frac{\max\{2130\sigma^2, 72G\sigma\}\log(4/\delta)}{T}$$

$$+ (\rho + L)\left(2130(\log(4/\delta))^{2\theta}\sigma^2 + 9G^2\right)\eta \quad (55)$$

Let we set $\eta = \sqrt{\frac{3\Delta_1}{(\rho+L)(2130(\log(4/\delta))^{2\theta}\sigma^2+9G^2)T}}$, then we have

$$\frac{1}{T}\sum_{t=1}^{T}\|\Theta(x_t)\|^2 \leq 2\sqrt{\frac{(\rho + L)\Delta_1\left(6390(\log(4/\delta))^{2\theta}\sigma^2 + 27G^2\right)}{T}}$$

$$+ \frac{\max\{2130\sigma^2, 72G\sigma\}\log(4/\delta)}{T} \quad (56)$$

- For $\theta > 1$, we get

$$\frac{1}{T}\sum_{t=1}^{T}\|\Theta(x_t)\|^2 \leq \frac{3\Delta_1}{T\eta} + + \frac{\hat{D}(\theta)\log(4/\delta)}{T} + (\rho + L)\left(18(11\theta\log(4/\delta))^{2\theta}\sigma^2 + 9G^2\right)\eta$$

$$(57)$$

Setting $\eta = \sqrt{\frac{3\Delta_1}{(\rho+L)(18(11\theta\log(4/\delta))^{2\theta}\sigma^2+9G^2)T}}$, then we have

$$\frac{1}{T}\sum_{t=1}^{T}\|\Theta(x_t)\|^2 \leq 2\sqrt{\frac{(\rho + L)\Delta_1\left(54(11\theta\log(4/\delta))^{2\theta}\sigma^2 + 27G^2\right)}{T}} + \frac{\hat{D}(\theta)\log(4/\delta)}{T}$$

$$(58)$$

$\square$

## D    PROOF SKETCH OF THEOREM 4.2

Here we briefly summarize the high-probability convergence proof process of the Riemannian clipped-SsGD method for weakly convex optimization under $p$-th moment noise. The definitions of $\mathcal{F}_t$ and $\mathbb{E}_t$ are the same as those in Section B. Our analysis again begins with Lemma B.2:

$$\frac{\bar{\rho} - \rho - L}{\bar{\rho}}\eta_t\|\Theta(x_t)\|^2 \leq \Delta_t - \Delta_{t+1} + \bar{\rho}\eta_t\langle \hat{x}_t - x_t, \xi_t\rangle + \bar{\rho}\eta_t^2\|\xi_t\|^2 + \bar{\rho}\eta_t^2 G^2 \quad (59)$$

Note that $\partial_t$ is defined such that $\mathbb{E}_t[\overline{\tilde{\nabla}_{\mathcal{R}} f(x_t)}] \in \partial_{\mathcal{R}} f(x_t)$ and let $\xi_t \triangleq \tilde{\nabla}_{\mathcal{R}} f(x_t) - \partial_t$. In contrast to the unclipped Riemannian SsGD method, $\tilde{\nabla}_{\mathcal{R}} f(x_t)$ here represents a clipped Riemannian stochastic subgradient, implying that $\tilde{\nabla}_{\mathcal{R}} f(x_t)$ is no longer an unbiased estimator of the subgradient of function $f$ at $x_t$, that is, $\mathbb{E}_t[\tilde{\nabla}_{\mathcal{R}} f(x_t)] \notin \partial_{\mathcal{R}} f(x_t)$. Consequently, the term $\bar{\rho}\eta_t\langle \hat{x}_t - x_t, \xi_t\rangle$ does not form a martingale, which complicates the high-probability upper-bound analysis of its summation. To overcome this challenge, we decompose $\xi_t$ into $\xi_t^u$ and $\xi_t^b$, where $\xi_t^u \triangleq g_t - \mathbb{E}_t[g_t]$ is the unbiased component and $\xi_t^b \triangleq \mathbb{E}_t[g_t] - \partial_t$ is the biased component. After this decomposition, inequality (59) becomes:

$$\frac{\bar{\rho} - \rho - L}{\bar{\rho}}\eta_t\|\Theta(x_t)\|^2 \leq \Delta_t - \Delta_{t+1} + \bar{\rho}\eta_t\langle \hat{x}_t - x_t, \xi_t^u\rangle + \bar{\rho}\eta_t\langle \hat{x}_t - x_t, \xi_t^b\rangle$$
$$+ \bar{\rho}\eta_t^2(2\|\xi_t^u\|^2 + 2\|\xi_t^b\|^2 + G^2) \quad (60)$$

For the subsequent analysis, we introduce the following lemma.

**Lemma D.1.** *Let $\lambda_t \geq 2G$, then for any $t \in [T]$, we have*

$$\mathbb{E}_t[\|\xi_t\|^2], \mathbb{E}_t[\|\xi_t^u\|^2], \|\xi_t^b\|^2 \leq \frac{10(2 - B^{-1})\sigma^p\lambda_t^{2-p}}{B^{p-1}},$$
$$\|\xi_t^u\| \leq 2\lambda_t, \quad \|\xi_t^b\| \leq \frac{2(2 - B^{-1})\sigma^p\lambda_t^{1-p}}{B^{p-1}} \quad (61)$$

Lemma D.1 is a direct Riemannian extension of Lemma 4 in Zhu et al. (2025). Similar results can be found in Zhang et al. (2020), Gorbunov et al. (2020), Nguyen et al. (2023), Liu et al. (2023b), Liu & Zhou (2023b), Liu et al. (2023a) and Liu et al. (2024), where the batch size is 1. By Lemma D.1, increasing $\lambda_t$ will result in an increase in the norm of $\xi_t^u$ and a decrease in the norm of $\xi_t^b$. This is because $\|\xi_t^b\| \leq 4\sigma^p\lambda_t^{1-p}/B^{p-1}$, and for all $p \in (1, 2]$, it holds that $1 - p \leq 2$. This observation prompts us to consider how to select an appropriate $\lambda_t$ to balance the orders of $\|\xi_t^u\|$ and $\|\xi_t^b\|$. Based on the hint in Liu & Zhou (2023b) and the inspiration in Zhu et al. (2025), we can similarly set an upper bound for $\eta_t\lambda_t$, that is, $\eta_t\lambda_t \leq \eta_0$.

Before proving Lemma D.3 and Lemma D.4, we first present the following lemma, which can be directly derived from our choices of $\eta_t$ and $\lambda_t$.

**Lemma D.2.** *If we set $\lambda_t = \max\{2G, \lambda t^{1/p}\}$, $\eta_t = \eta_0 \min\{1/\lambda_t, 1/(G\sqrt{t})\}$, where $\lambda$ and $\eta_0$ are arbitrary positive constants, the following inequalities hold:*

$$\eta_t\lambda_t \leq \eta_0, \quad (\sigma/\lambda_t)^p \leq (\sigma/\lambda)^p t^{-1}, \quad G^2\eta_t^2 \leq \eta_0^2 t^{-1} \quad (62)$$

After decomposing $\xi$ into $\xi_t^u$ and $\xi_t^b$, it is easy to observe that $\bar{\eta}_t\langle \hat{x}_t - x_t, \xi_t^u\rangle$ is indeed a martingale, which means we can once again apply the concentration inequality to bound the summation. Therefore, we can obtain the following lemma.

**Lemma D.3.** *We define $\lambda_t = \max\{\lambda t^{1/p}, 2G\}$ and require $\eta_t \leq \eta_0/\lambda_t$. With $\bar{\rho} = 2(\rho + L)$, the following inequality holds with probability at least $1 - \delta/2$:*

$$\sum_{t=1}^{T} \bar{\rho}\eta_t\langle \hat{x}_t - x_t, \xi_t^u\rangle = O\left(G\eta_0\left(\log(1/\delta) + \sqrt{(\sigma/\lambda)^p B^{1-p}\log(1/\delta)\log T}\right)\right) \quad (63)$$

*Proof.* First of all, we notice that

$$|\bar{\rho}\eta_t\langle \hat{x}_t - x_t, \xi_t^u\rangle| \leq \bar{\rho}\eta_t\|\hat{x}_t - x_t\|\|\xi_t^u\| \leq \bar{\rho}\eta_t \cdot \frac{2G}{\bar{\rho} - \rho - L} \cdot 2\lambda_t \leq 8G \cdot \lambda_t\eta_t \leq 8G\eta_0 \quad (64)$$

where the second inequality follows from Lemma B.4 and B.3 and the last inequality is derived using Lemma D.2. We also get

$$
\begin{aligned}
\mathbb{E}_t[|\bar{\rho}\eta_t\langle \hat{x}_t - x_t, \xi_t^u\rangle|^2] &\leq 4(\rho+L)^2\eta_t^2\|\hat{x}_t - x_t\|^2\mathbb{E}[\|\xi_t^u\|^2] \\
&\leq 160G^2(\eta_t\lambda_t)^2 \cdot \frac{(2-B^{-1})(\sigma/\lambda_t)^p}{B^{p-1}} \\
&\leq 320(G\eta_0)^2 B^{1-p}(\sigma/\lambda)^p \cdot \frac{1}{t}
\end{aligned}
\tag{65}
$$

Therefore, we can obtain

$$
\sum_{t=1}^T \mathbb{E}_t[|\bar{\rho}\eta_t\langle \hat{x}_t - x_t, \xi_t^u\rangle|^2] \leq 320(G\eta_0)^2 B^{1-p}(\sigma/\lambda)^p \cdot \log(eT)
\tag{66}
$$

To use Lemma A.2, set $R = 8G\eta_0$ and $F = 320(G\eta_0)^2 B^{1-p}(\sigma/\lambda)^p \cdot \log(eT)$, Then, the following inequality holds with the probability at least $1 - \delta/2$:

$$
\begin{aligned}
\sum_{t=1}^T |\bar{\rho}\eta_t\langle \hat{x}_t - x_t, \xi_t^u\rangle| &\leq \frac{2R}{3}\log(4/\delta) + \sqrt{2F\log(4/\delta)} \\
&\leq \frac{16\eta_0 G\log(4/\delta)}{3} + \sqrt{640B^{1-p}(G\eta_0)^2(\sigma/\lambda)^p\log(eT)\log(4/\delta)} \\
&\leq 26G\eta_0\left(\log(4/\delta) + \sqrt{B^{1-p}(\sigma/\lambda)^p\log(eT)\log(4/\delta)}\right)
\end{aligned}
\tag{67}
$$

$\square$

The current challenge lies in limiting the summation of $\eta_t^2\|\xi_t^u\|^2$. By using Lemma D.1, we can directly bound it using $4\lambda_t^2$. However, applying the bound, we have $\eta_t^2\lambda_t^2 \leq \eta_0^2$, which is not sufficient to control the summation. Therefore, we can divide $\|\xi_t^u\|^2$ into two terms, namely $\|\xi_t^u\|^2 - \mathbb{E}_t[\|\xi_t^u\|^2]$ and $\mathbb{E}_t[\|\xi_t^u\|^2]$. Note that $\|\xi_t^u\|^2 - \mathbb{E}_t[\|\xi_t^u\|^2]$ is a martingale. Regarding the summation of this item, we can obtain the following lemma.

**Lemma D.4.** *We define* $\lambda_t = \max\{\lambda t^{1/p}, 2G\}$ *and require* $\eta_t \leq \eta_0/\lambda_t$. *With* $\bar{\rho} = 2(\rho+L)$, *the following inequality holds with probability at least* $1 - \delta/2$:

$$
\sum_{t=1}^T \bar{\rho}\eta_t^2(\|\xi_t^u\|^2 - \mathbb{E}_t[\|\xi_t^u\|^2]) = O\left(\rho\eta_0^2\left(\log(1/\delta) + \sqrt{(\sigma/\lambda)^p B^{1-p}\log(1/\delta)\log T}\right)\right)
\tag{68}
$$

*Proof.* Firstly, we get

$$
\left|\bar{\rho}\eta_t^2(\|\xi_t^u\|^2 - \mathbb{E}_t[\|\xi_t^u\|^2])\right| \leq \bar{\rho}\eta_t^2(\|\xi_t^u\|^2 + \mathbb{E}_t[\|\xi_t^u\|^2]) \leq 8\bar{\rho}(\lambda_t\eta_t)^2 \leq 16(\rho+L)\eta_0^2
\tag{69}
$$

Then we have

$$
\begin{aligned}
\mathbb{E}_t&\left[\left|\bar{\rho}\eta_t^2(\|\xi_t^u\|^2 - \mathbb{E}_t[\|\xi_t^u\|^2])\right|^2\right] \\
&= \mathbb{E}_t\left[\bar{\rho}^2\eta_t^4\left(\|\xi_t^u\|^4 - 2\langle\|\xi_t^u\|^2, \mathbb{E}_t[\|\xi_t^u\|^2]\rangle + (\mathbb{E}_t[\|\xi_t^u\|^2])^2\right)\right] \\
&\leq \bar{\rho}^2\eta_t^4\left(\mathbb{E}_t[\|\xi_t^u\|^4] - 2(\mathbb{E}_t[\|\xi_t^u\|^2])^2 + (\mathbb{E}_t[\|\xi_t^u\|^2])^2\right) \\
&\leq \bar{\rho}^2\eta_t^4\mathbb{E}_t[\|\xi_t^u\|^4] \leq \bar{\rho}^2\eta_t^4 \cdot 4\lambda_t^2\mathbb{E}_t[\|\xi_t^u\|^2] \\
&\leq 4\bar{\rho}^2\eta_0^2(\eta_t\lambda_t)^2 \cdot 20(\sigma/\lambda_t)^p B^{1-p} \\
&\leq 320(\rho+L)^2\eta_0^4(\sigma/\lambda)^p B^{1-p}t^{-1}
\end{aligned}
\tag{70}
$$

Furthermore, we have a upper bound

$$
\sum_{t=1}^T \mathbb{E}_t\left[\left|\bar{\rho}\eta_t^2(\|\xi_t^u\|^2 - \mathbb{E}_t[\|\xi_t^u\|^2])\right|^2\right] \leq 320(\rho+L)^2\eta_0^4(\sigma/\lambda)^p B^{1-p}\log(eT)
\tag{71}
$$

Similarly, we set $R = 16(\rho + L)\eta_0^2$ and $F = 320(\rho + L)^2\eta_0^4(\sigma/\lambda)^p B^{1-p}\log(eT)$ then use Lemma A.2 and have the following inequality holds with the probability at least $1 - \delta/2$:

$$
\sum_{t=1}^{T} |\bar{\rho}\eta_t^2(\|\xi_t^u\|^2 - \mathbb{E}_t[\|\xi_t^u\|^2])|
$$

$$
\leq \frac{2R\log(4/\delta)}{3} + \sqrt{2F\log(4/\delta)}
$$

$$
\leq \frac{32}{3}(\rho + L)\eta_0^2\log(4/\delta) + \sqrt{640(\rho + L)^2\eta_0^4(\sigma/\lambda)^p\log(eT)\log(4/\delta)B^{1-p}}
$$

$$
\leq 26(\rho + L)\eta_0^2\left(\log(4/\delta) + \sqrt{(\sigma/\lambda)^p\log(eT)\log(4/\delta)B^{1-p}}\right) \tag{72}
$$

$\square$

Based on Lemmas D.3 and D.4, the growth rates for the summations of both $\bar{\rho}\eta_t\langle \hat{x}_t - x_t, \xi_t^u\rangle$ and $\bar{\rho}\eta_t^2(\|\xi_t^u\|^2 - \mathbb{E}_t[\|\xi_t^u\|^2])$ can be bounded by $O(\sqrt{\log T})$ with probability at least $1 - \delta$. Regarding the term $\bar{\rho}\eta_t\langle \hat{x}_t - x_t, \xi_t^b\rangle$, we can observe from Appendix E.1 the total summation of $\bar{\rho}\eta_t\langle \hat{x}_t - x_t, \xi_t^b\rangle$ be upper-bounded by $O(\log T)$.

The remaining terms, $\bar{\rho}\eta_t\left(2\mathbb{E}_t[\|\xi_t^u\|^2 + 2\|\xi_t^b\|^2 + \|\partial_t\|^2]\right)$, can similarly be bounded in summation by $O(\log T)$ using similar techniques. Furthermore, to bound the summation of $\eta_t^2\|\partial_t\|^2$, we impose the additional condition $\eta_t \leq \eta_0/(G\sqrt{t})$.

# E   RIEMANNIAN PROJECTED CLIPPED-SSGD UNDER THE $p$-BCM NOISES

## E.1   PROOF OF THEOREM 4.2

*Proof.* Set $\bar{\rho} = 2(\rho + L)$ and sum both sides of (60) from 1 to $T$, which yields

$$
\frac{1}{2}\sum_{t=1}^{T}\eta_t\|\Theta(x_t)\|^2 \leq \Delta_1 + \sum_{t=1}^{T}2(\rho + L)\eta_t\langle \hat{x}_t - x_t, \xi_t^u\rangle + \sum_{t=1}^{T}4(\rho + L)\eta_t^2(\|\xi_t^u\|^2 - \mathbb{E}_t[\|\xi_t^u\|^2])
$$

$$
+ \sum_{t=1}^{T}2(\rho + L)\eta_t\langle \hat{x}_t - x_t, \xi_t^b\rangle + \sum_{t=1}^{T}2(\rho + L)\eta_t^2(2\mathbb{E}_t[\|\xi_t^u\|^2] + 2\|\xi_t^b\|^2 + G^2)
$$

$$
\tag{73}
$$

By combining Lemma D.3 and Lemma D.4, we obtain the following inequality with the probability at least $1 - \delta$:

$$
\sum_{t=1}^{T}2(\rho + L)\eta_t\langle \hat{x}_t - x_t, \xi_t^u\rangle + \sum_{t=1}^{T}4(\rho + L)\eta_t^2(\|\xi_t^u\|^2 - \mathbb{E}_t[\|\xi_t^u\|^2])
$$

$$
\leq \left(26G\eta_0 + 52(\rho + L)\eta_0^2\right)\left(\log(4/\delta) + \sqrt{(\sigma/\lambda)^p\log(eT)\log(4/\delta)B^{1-p}}\right)
$$

$$
\leq \left(26G\eta_0 + 52(\rho + L)\eta_0^2\right)\left(\frac{3}{2}\log(4/\delta) + \frac{(\sigma/\lambda)^p\log(eT)}{2B^{p-1}}\right)
$$

$$
\tag{74}
$$

For the remaining terms, the following deterministic bounds hold:

$$
\sum_{t=1}^{T}2(\rho + L)\eta_t\langle \hat{x}_t - x_t, \xi_t^b\rangle \leq \sum_{t=1}^{T}2(\rho + L)\eta_t\|\hat{x}_t - x_t\|\|\xi_t^b\|
$$

$$
\leq \sum_{t=1}^{T}4G\cdot(\eta_t\lambda_t)\cdot\frac{2(2 - B^{-1})(\sigma/\lambda_t)^p}{B^{p-1}}
$$

$$
\leq 16G\eta_0(\sigma/\lambda)^p B^{1-p}\sum_{t=1}^{T}t^{-1}
$$

$$
\leq 16G\eta_0(\sigma/\lambda)^p B^{1-p}\log(eT) \tag{75}
$$

and

$$2(\rho + L)\sum_{t=1}^{T}\eta_t^2(2\mathbb{E}_t\left\|\xi_t^u\right\|^2 + 2\left\|\xi_t^b\right\|^2 + G^2)$$

$$\leq 2(\rho + L)\sum_{t=1}^{T}\eta_t^2\left(40(2 - B^{-1})\sigma^p\lambda_t^{2-p}B^{1-p} + G^2\right)$$

$$\leq \sum_{t=1}^{T}160(\rho + L)(\eta_t\lambda_t)^2(\sigma/\lambda_t)^pB^{1-p} + \sum_{t=1}^{T}2(\rho + L)G^2\eta_t^2$$

$$\leq 160(\rho + L)\eta_0^2(\sigma/\lambda)^pB^{1-p}\log(eT) + 2(\rho + L)\eta_0^2\log(eT) \quad (76)$$

By substituting (74), (75), and (76) into (73), we obtain the following inequality, which holds with probability at least $1 - \delta$:

$$\frac{1}{2}\sum_{t=1}^{T}\eta_t\left\|\Theta(x_t)\right\|^2 \leq \Delta_1 + \left(29G\eta_0 + 186(\rho + L)\eta_0^2\right)(\sigma/\lambda)^pB^{1-p}\log(eT)$$

$$+ \left(39G\eta_0 + 78(\rho + L)\eta_0^2\right)\log(4/\delta) + 2(\rho + L)\eta_0^2\log(eT)$$

$$\leq \Delta_1 + \left(39G\eta_0 + 186(\rho + L)\eta_0^2\right)\left((\sigma/\lambda)^pB^{1-p}\log(eT) + \log(4/\delta)\right)$$

$$+ 2(\rho + L)\eta_0^2\log(eT) \quad (77)$$

Multiplying both sides of the above inequality by $2/(\eta_T T) = (2/\eta_0)\max\{\lambda_T/T, G/\sqrt{T}\} \leq 2/\eta_0\max\{\lambda/T^{(p-1)/p}, G/\sqrt{T}\}$, then we obtain

$$\frac{1}{T}\sum_{t=1}^{T}\left\|\Theta(x_t)\right\|^2 \leq \left\{\frac{2\Delta_1}{\eta_0} + (78G + 372(\rho + L)\eta_0)\left((\sigma/\lambda)^pB^{1-p}\log(eT) + \log(4/\delta)\right)\right.$$

$$\left. + 4(\rho + L)\eta_0\log(eT)\right\}\max\left\{\frac{\lambda}{T^{(p-1)/p}}, \frac{G}{\sqrt{T}}\right\} \quad (78)$$

If $\lambda = \sigma$ and $B = 1$, then we can get

$$\frac{1}{T}\sum_{t=1}^{T}\left\|\Theta(x_t)\right\|^2 \leq \left\{\frac{2\Delta_1}{\eta_0} + (78G + 376(\rho + L)\eta_0)\log(4eT/\delta)\right\}\cdot\max\left\{\frac{\lambda}{T^{(p-1)/p}}, \frac{G}{\sqrt{T}}\right\} \quad (79)$$

$\square$

### E.2 PROOF OF THEOREM 4.4

*Proof.* Taking the expectation on both sides of inequality 73, we can obtain

$$\frac{1}{2}\sum_{t=1}^{T}\eta_t\mathbb{E}\left[\left\|\Theta(x_t)\right\|^2\right] \leq \Delta_1 + \sum_{t=1}^{T}2(\rho + L)\eta_t\mathbb{E}\langle\hat{x}_t - x_t, \xi_t^b\rangle$$

$$+ \sum_{t=1}^{T}2(\rho + L)\eta_t^2\left(2\mathbb{E}\|\xi_t^u\|^2 + 2\mathbb{E}\|\xi_t^b\|^2 + G^2\right)$$

$$\leq \Delta_1 + \sum_{t=1}^{T}2(\rho + L)\eta_t\mathbb{E}\left[\|\hat{x}_t - x_t\|\cdot\|\xi_t^b\|\right]$$

$$+ \sum_{t=1}^{T}2(\rho + L)\eta_t^2\mathbb{E}\left[2\mathbb{E}_t\|\xi_t^u\|^2 + 2\|\xi_t^b\|^2 + G^2\right]$$

$$\leq \Delta_1 + 16G\eta_0(\sigma/\lambda)^pB^{1-p}\log(eT)$$

$$+ 160(\rho + L)\eta_0^2(\sigma/\lambda)^pB^{1-p}\log(eT) + 2(\rho + L)\eta_0^2\log(eT)$$

$$= \Delta_1 + \left(16G\eta_0 + 160(\rho + L)\eta_0^2\right)(\sigma/\lambda)^pB^{1-p}\log(eT)$$

$$+ 2(\rho + L)\eta_0^2\log(eT) \quad (80)$$

Multiplying both sides of the above inequality by $2/(\eta_T T) = (2/\eta_0) \max\{\lambda_T/T, G/\sqrt{T}\} \leq 2/\eta_0 \max\{\lambda/T^{(p-1)/p}, G/\sqrt{T}\}$, then we obtain

$$\frac{1}{T} \sum_{t=1}^{T} \mathbb{E}\left[\|\Theta(x_t)\|^2\right] \leq \left\{\frac{2\Delta_1}{\eta_0} + (32G + 320(\rho+L)\eta_0)(\sigma/\lambda)^p B^{1-p} \log(eT)\right.$$
$$\left. + 4(\rho+L)\eta_0 \log(eT)\right\} \cdot \max\left\{\frac{\lambda}{T^{(p-1)/p}}, \frac{G}{\sqrt{T}}\right\} \tag{81}$$

If $\lambda = \sigma$ and $B = 1$, then we can have

$$\frac{1}{T} \sum_{t=1}^{T} \mathbb{E}\left[\|\Theta(x_t)\|^2\right] \leq \left\{\frac{2\Delta_1}{\eta_0} + (32G + 324(\rho+L)\eta_0) \log(eT)\right\} \cdot \max\left\{\frac{\lambda}{T^{(p-1)/p}}, \frac{G}{\sqrt{T}}\right\}$$
$$\tag{82}$$

$\square$

