# OpenReview forum: "Riemannian Stochastic Weakly Convex Optimization Under Heavy-Tailed Noises"
_ICLR.cc/2026/Conference — Submitted to ICLR 2026_

### Official Review · Reviewer_3teB · 2025-10-25

**Soundness:** 3
**Presentation:** 2
**Contribution:** 2
**Rating:** 4
**Confidence:** 4

**Summary:**

The paper extends the convergence analysis proposed by Zhu et al. (2025) (Euclidean space) to optimization problems on Stiefel manifolds. The main contributions of this work are summarized as follows:

1)	For Riemannian SsGD (Riemannian Stochastic Subgradient Descent) on the Stiefel manifold, the authors derive the first high-probability convergence rate. This result is established under the setting of nonsmooth weakly convex optimization with sub-Weibull type noise.

2)	For mini-batch Riemannian clipped SsGD on the Stiefel manifold, the authors also obtain the first high-probability convergence rate. This finding applies to nonsmooth weakly convex optimization with p-BCM type noise.

Tianxi Zhu, Yi Xu, and Xiangyang Ji. Stochastic weakly convex optimization under heavy-tailed noises. arXiv preprint arXiv:2507.13283, 2025.

**Strengths:**

The authors extend heavy-tailed optimization to Stiefel manifolds and provide a theoretical guarantee for Riemannian stochastic algorithms in some real-world scenarios.

**Weaknesses:**

1.	The definition should be presented before the term is used, such as σ-sub-Weibull(θ) distribution and p-BCM condition.

2.	Lack of experimental validation. The main test does not include numerical experiments. To verify the theoretical results, supplementary experiments on the Stiefel manifold under heavy-tailed noise conditions are needed. Additionally, it would be valuable to demonstrate whether the derived result constitutes the tightest possible upper bound.

3.	Limited manifold generalization. The work only focuses on the Stiefel manifold but does not discuss adaptation to other common Riemannian manifolds. It would be valuable to analyze how manifold geometric properties affect the algorithm’s convergence.

4.	Insufficient comparison with related work. Given that the authors mention in the abstract that “the dependence on failure probability and iteration complexity matches the best-known Euclidean results up to constants,” it is essential to explicitly highlight the key differences between the proposed work and existing best-known related results in the paper.


Minor:

1.	Equation(17) should be \(E[||x||^p]\leq \sigma^p <+\infity\)

2.	Equation(39) should be \(E_t[exp\{(||\xi_t||/\sigma)^\frac{1}{\theta}\}]\)

**Questions:**

The paper’s title is “RIEMANNIAN STOCHASTIC WEAKLY CONVEX OPTIMIZATION UNDER HEAVY-TAILED NOISES”. Given this focus on Riemannian optimization, two key questions arise:

1.	Could this work be extended to the common Riemannian manifolds? The paper only focuses on optimization problems on the Stiefel manifolds.

2.	How do the geometric properties of the manifold influence the algorithm’s convergence, compared with those in Euclidean space?

---

### Official Review · Reviewer_kERU · 2025-10-25

**Soundness:** 2
**Presentation:** 2
**Contribution:** 2
**Rating:** 4
**Confidence:** 4

**Summary:**

This paper considers the stochastic weakly convex opt over Stiefel manifolds. Assuming the stochastic gradients are unbiased and with heavy-tailed noises, the authors design Riemannian stochastic subgradient methods and clipped-type methods with theoretical guarantees.

**Strengths:**

It's interesting to consider the heavy-talied noise setting in the Riemannian weakly cnvex setting.

**Weaknesses:**

1. It is not entirely convincing that the setting (i.e., Riemannian weakly convex opt with heavy-tailed noises on the stochastic gradients) makes sense. Could the authors showcase its practical relevance by giving (several) concrete examples? The only example briefly mentioned in line 58 is not about Stiefel manifolds. Perhaps the the heavy-tailed noises appear in distributed settings (https://arxiv.org/abs/2303.17779) but I am not sure.
2. When going through this paper, I did not grasp the key technical difficulty compared with the Euclidean case. Could the authors explicitly point out the key technical difficulty?
3. Line 75 and Line 236, "weak" should be "weakly". Line 149, "close" should be "closed".

**Questions:**

See the two questions in the above weakness part.

---

### Official Review · Reviewer_eHbK · 2025-11-01

**Soundness:** 3
**Presentation:** 3
**Contribution:** 2
**Rating:** 4
**Confidence:** 3

**Summary:**

The paper studies the Riemannian SGD in a heavy-tailed noise setup. The authors propose two algorithms, that handle sub-Weibull and $p$
-BCM noises. They derive convergence rates both in expectation and with high-probability. The results obtained do not differ significantly from the ones derived for the Euclidean setup in existing works.

**Strengths:**

1)Sub-Weibull random variables, as well as random variables with bounded $p$-th central moment, are considered.

2)General convergence rates for $\eta_t$, satisfying condition at line 241, are derived

**Weaknesses:**

1)Only Stiefel manifolds are considered.

2)Even though the authors require weak convexity, convergence results are given for $\Theta(x)$, instead of the function's value.

3)No experiments, analyzing the performance of Algorithm 1 and Algorithm 2.

**Questions:**

1)What is the motivation for considering the weakly-convex setup for Riemannian optimization?

2)Can you provide experiments, validating, that your derived methods handle heavy-tailed noise for Riemannian setup?

3)How can the assumptions on gradient's boundedness be relaxed to the smoothness?

**Details Of Ethics Concerns:**

No additional ethical concerns.

---

### Official Review · Reviewer_eu7t · 2025-11-03

**Soundness:** 2
**Presentation:** 2
**Contribution:** 2
**Rating:** 4
**Confidence:** 3

**Summary:**

Extend the convergence analysis of weakly convex optimization under heavy detail noise from Euclidean space to Stiefel manifold.

**Strengths:**

Analysis of SGD on manifold under heavy tail noise.

**Weaknesses:**

It seems the extension is fully parallel with the Euclidean case, and some numerical experiments may be helpful.

**Questions:**

* What is the key challenge for the extension from the Euclidean space to Stiefel manifold?
* The definition of retraction should be properly introduced. In fact, what property should the retraction satisfy in order to establish the convergence? I see from the proof of Lemma B.2 that the retraction based on the polar decomposition is used. Why and whether does the result hold for other retractions?
* Can the analysis be extended beyond Stiefel manifold?
* Assumption 3.5 only assumes $\sigma$-sub-Weilbull($\theta$), but the $p$-BCM noise is also considered.
* Any example to show that Assumption 3.3 can be met?
* Is it possible to obtain strong result for example if $f$ is (locally) geodesic strongly convex?

---

### Meta-Review · Area_Chair_HB2w · 2026-01-07

**Summary:**

This paper presents a theoretical extension of stochastic optimization to the Riemannian setting under heavy-tailed noise. The reviewers were unanimous in their assessment (all giving a score of 4) that the submission falls short on two critical fronts: lack of empirical validation and incremental novelty from Euclidean results. Some reviewers also questioned the practical relevance of the specific setting (weakly convex on Stiefel with heavy tails).

**Reviewer Concerns:**

All concerns remain outstanding, as the authors did not participate in the rebuttal process.

**Reviewer Scores:**

The reviewers likely would have maintained or lowered their scores.

---

### Decision · Program_Chairs · 2026-01-26

Reject